# A Comprehensive Characterization of Empirical Parameterizations for OH Exposure in the Aerodyne Potential Aerosol Mass Oxidation Flow Reactor (PAM-OFR)

Qianying Liu[1,2], Dan Dan Huang[2,*], Andrew T. Lambe[3], Shengrong Lou[2], Lulu Zeng[1], Yuhang Wu[2], Congyan Huang[2], Shikang Tao[2], Xi Cheng[4], Qi Chen[5], Ka In Hoi[1], Hongli Wang[2], Kai Meng Mok[1], Cheng Huang[2,6], Yong Jie Li[1,*]

[1]Department of Civil and Environmental Engineering, Department of Ocean Science and Technology, and Centre for Regional Oceans, Faculty of Science and Technology, University of Macau, Taipa, Macau SAR, 999078, China
[2]State Environmental Protection Key Laboratory of Formation and Prevention of Urban Air Pollution Complex, Shanghai Academy of Environmental Sciences, Shanghai, 200233, China
[3]Aerodyne Research Inc., Billerica, Massachusetts, 01821, United States
[4]School of Chemical and Environmental Engineering, China University of Mining and Technology (Beijing), Beijing 100083, China
[5]State Key Joint Laboratory of Environmental Simulation and Pollution Control, BIC-ESAT and IJRC, College of Environmental Sciences and Engineering, Peking University, Beijing, China
[6]State Ecology and Environment Scientific Observation and Research Station for the Yangtze River Delta at Dianshan Lake ,Shanghai Environmental Monitoring Center, Shanghai, 200030, China

*Correspondence to*: Dan Dan Huang (huangdd@saes.sh.cn), Yong Jie Li (yongjieli@um.edu.mo)

**Abstract.** The oxidation flow reactor (OFR) has been widely used to simulate secondary organic aerosol (SOA) formation in laboratory and field studies. OH exposure ($OH_{exp}$), representing the extent of hydroxyl radical (OH) oxidation and normally expressed as the product of OH concentration and residence time in the OFR, is important in assessing the oxidation chemistry in SOA formation. Several models have been developed to quantify the $OH_{exp}$ in OFRs, and empirical equations have been proposed to parameterize $OH_{exp}$. Practically, the empirical equations and the associated parameters are derived under atmospheric relevant conditions (i.e., external OH reactivity) with limited variations of calibration conditions, such as residence time, water vapor mixing ratio, $O_3$ concentration, etc. Whether the equations or parameters derived under limited sets of calibration conditions can accurately predict the $OH_{exp}$ under dynamically changing experimental conditions with large variations (i.e., extremely high external OH reactivity) in real applications remains uncertain. In this study, we conducted 62 sets of experiments (416 data points) under a wide range of experimental conditions to evaluate the scope of the application of the empirical equations to estimate $OH_{exp}$. Sensitivity tests were also conducted to obtain a minimum number of data points that is necessary for generating the fitting parameters. We showed that, for the OFR185 mode (185-nm lamps with internal $O_3$ generation), except for external OH reactivity, the parameters obtained within a narrow range of calibration conditions can be extended to estimate the $OH_{exp}$ when the experiments are in wider ranges of conditions. For example, parameters derived within a narrow water vapor mixing ratio range (0.49–0.99 %, corresponding to 15.1–30.8 % of relative humidity at 101.325 kPa and 298 K) can be extended to estimate the $OH_{exp}$ under the entire range of water vapor mixing ratios (0.49–2.76 %,

equivalent to 15.1–85.7 % of relative humidity under identical conditions). However, the parameters obtained when the
external OH reactivity is below 23 s⁻¹ could not be used to reproduce the $OH_{exp}$ under the entire range of external OH reactivity
(4–204 s⁻¹). For the OFR254 mode (254-nm lamps with external $O_3$ generation), all parameters obtained within a narrow range
of conditions can be used to estimate $OH_{exp}$ accurately when experimental conditions are extended. Additionally, when using
the OFR254 mode, too-low lamp voltages should be avoided, as they will generally result in large deviations in the estimations
of $OH_{exp}$ from empirical equations. Regardless of OFR185 or OFR254 mode, at least 20–30 data points from sulfur dioxide
($SO_2$) or (carbon monoxide) CO decay with varying conditions are required to fit a set of empirical parameters that can
accurately estimate $OH_{exp}$. Caution should be exercised to use fitted parameters from low external OH reactivity to high ones,
for instance, those from direct emissions such as vehicular exhaust and biomass burning.
**1 Introduction**
As the most important oxidant in tropospheric chemistry (Ehhalt, 1999), hydroxyl (OH) radical is vital in oxidizing primary
pollutants such as volatile organic compounds (VOCs) and contributes to secondary organic aerosol (SOA) and tropospheric
ozone ($O_3$) formation. The OH radical has daytime concentrations of $10^5$ to $10^7$ molecules cm⁻³, exhibiting daily (Cao et al.,
2020; Tan et al., 2017), seasonal (Friedman and Farmer, 2018), as well as spatial (Cao et al., 2020; Stone et al., 2012) variations.
An average daily OH radical concentration of $1.5 \times 10^6$ molecules cm⁻³ is widely used to estimate the photochemical age of an
air mass (Mao et al., 2009). Typical VOCs have second-order rate constants of $10^{-15}$ to $10^{-10}$ cm³ molecule⁻¹ s⁻¹ with OH radicals
(Atkinson and Arey, 2003; Atkinson et al., 2006), which can be translated to atmospheric lifetimes of hours to approximately
a year (Seinfeld and Pandis, 2016). This situation poses challenges in laboratory experiments to directly simulate the OH
oxidation of VOCs, which is one of the most important chemical processes in the Earth's atmosphere.
Smog chambers (Cocker et al., 2001; Hildebrandt et al., 2009; Wang et al., 2014) and oxidation flow reactors (OFRs) (George
et al., 2007; Kang et al., 2007; Lambe et al., 2011) have been widely employed to simulate oxidation of VOCs and subsequent
SOA formation. For example, the Caltech Chamber provides oxidation conditions close to the real atmosphere, making it
suitable for the study of complex multi-step reactions and low-volatility products. However, each experiment takes several
hours to days and long-duration experiments are prone to background interference. The Toronto Photo-Oxidation Tube (TPOT)
focuses on the study of heterogeneous oxidation reactions of aerosols. Its 0.8 L volume makes it portable, but it is prone to
uneven residence time distribution (RTD) and significant wall effects. The Potential Aerosol Mass Oxidation Flow Reactor
(PAM-OFR) and the Gothenburg Potential Aerosol Mass Oxidation Flow Reactor (Go: PAM-OFR) are often used to study
the transformation of gaseous precursors into particles (such as the formation of SOA). The Go: PAM-OFR has a volume of
7.2 L, which is only half that of the PAM-OFR, making it suitable for experiments on mobile platforms. However, its small
volume gives it the same disadvantages as the TPOT, and it is equipped with only a single UV lamp, which does not allow for
as wide a range of controllable oxidation levels as the PAM-OFR. The PAM's moderate volume and central flow sampling can
reduce wall effects.
These OFRs of reactors normally operate with high concentrations of oxidants (e.g., OH radicals), which lead to a significant
acceleration of oxidation reactions, often by orders of magnitude. To reconcile the differences in OH concentration and
exposure time between ambient and laboratory settings, the oxidation extent, i.e., OH exposure ($OH_{exp}$, molecules $cm^{-3}$ s) is
normally used to extrapolate laboratory findings to ambient conditions. Despite drawbacks such as possible altered reaction
mechanisms, this approach provides a quantitative assessment of the chemistry during OH oxidation in a reasonable time span
and achievable detection capability. The $OH_{exp}$ has a significant impact on the yield and product distribution during VOC
oxidation (Cheng et al., 2021; Cheng et al., 2024). Accurate measurement or estimation of the $OH_{exp}$ during laboratory
experiments, therefore, is the key to understanding the oxidation chemistry that can represent the ambient conditions. In this
study, we chose to further investigate the PAM-OFR to explore its $OH_{exp}$, as it offers moderate conditions in terms of
experiment time, deployment complexity, range of oxidation levels, and wall effects.
The Aerodyne Potential Aerosol Mass OFR (PAM-OFR) is one of the most widely used OFRs for studying SOA formation
and evolution (Zhang et al., 2024). It can achieve a wide range of atmospheric $OH_{exp}$ conditions within short residence times
on the order of minutes (Kang et al., 2007; Lambe et al., 2011). The PAM-OFR can be operated in a number of modes,
depending on 1) the wavelength of the ultraviolet (UV) light source, 2) the concentration of the externally generated $O_3$ (if
any), and 3) the injection of external precursor to generate $NO_x$ (= NO + $NO_2$) or other oxidants (e.g., nitrate radical or halogen
atoms) upon photolysis. The most widely used methods for OH generation include combined photolysis of $O_2$ and $H_2O$ at $\lambda$ =
185 nm plus photolysis of $O_3$ at $\lambda$ = 254 nm (OFR185; R1–R6) or photolysis of externally added $O_3$ at $\lambda$ = 254 nm (OFR254;
R5–R6) (Rowe et al., 2020):
$H_2O + hv_{185} \rightarrow H + OH$                                                                                    (R1)
$H + O_2 \rightarrow HO_2$                                                                                              (R2)
$O_2 + hv_{185} \rightarrow 2O(^3P)$                                                                                    (R3)
$O(^3P) + O_2 \rightarrow O_3$                                                                                          (R4)
$O_3 + hv_{254} \rightarrow O_2 + O(^1D)$                                                                               (R5)
$O(^1D) + H_2O \rightarrow 2OH$                                                                                         (R6)
To obtain the $OH_{exp}$ under these two modes in the PAM-OFR, one can perform decay experiments on trace gases such as $SO_2$
and CO, and fit the $OH_{exp}$ based on known second-order rate constants between OH radical and the trace gases, which is
defined as $OH_{exp, dec}$. Based on the results of the decay experiments, Li et al. (2015) and Peng et al. (2015) developed estimation
equations to parameterize $OH_{exp}$ as a function of easily measurable quantities, which is denoted as $OH_{exp, est}$. A set of parameters
($a$–$f$ and $x$–$z$, respectively) for the estimation equations of the OFR185 and OFR254 modes (see Sect. 2.3 for details) were
obtained by fitting the estimation equations to $OH_{exp, dec}$ values obtained from decay experiments.
When using the PAM-OFR in field studies, it is necessary to obtain concurrent $OH_{exp}$ that is representative of the ambient
conditions. However, environmental conditions in field studies (e.g., humidity, temperature, etc.) are constantly changing,
making it challenging to replicate these conditions for $OH_{exp}$ estimation. In some field studies using PAM-OFR, concurrent
$OH_{exp}$ was estimated by measuring the relative decay of benzene and toluene (Liao et al., 2021; Liu et al., 2018). Additionally,
some studies have mentioned that OH concentrations can be indirectly measured by detecting the decay of tracers such as 3-
pentanol, 3-pentanone, pinonaldehyde, or butanol-d9 (Barmet et al., 2012). However, the measurement of all these organic
tracers requires specific, sophisticated instruments such as proton-transfer-reaction time-of-flight mass spectrometers (PTR-
MS). Additionally, switching the instrument back and forth between the front and end of the OFR during field measurements
can result in some loss of real-time VOCs data before entering the OFR. To obtain accurate $OH_{exp}$, some studies explicitly
modelled the radical chemistry in PAM-OFR (Li et al., 2015; Ono et al., 2014; Peng et al., 2015). The estimation equations
developed by  Li et al. (2015) and Peng et al. (2015), although empirical, reproduced the $OH_{exp}$ from models within 10 %,
making them a good choice because these equations only require the input of a few easily available parameters. Yet, it is
unclear whether the fitted parameters obtained under certain conditions can still accurately estimate $OH_{exp}$ when experimental
conditions, such as UV light intensity, water vapor mixing ratio, residence time, and external OH reactivity ($OHR_{ext}$), undergo
significant changes. Furthermore, there is currently no consensus on the minimum number of decay experiments required to
obtain accurate parameterization for $OH_{exp}$ estimation using these equations. This facet is important for field studies using
PAM-OFR where only limited numbers of decay experiments can be done to obtain concurrent $OH_{exp}$ estimation.
In this study, we conducted a series of experiments using the decay of $SO_2$ and CO to estimate the $OH_{exp}$ in the PAM-OFR
under OFR185 and OFR254 modes. The applicability of previously developed $OH_{exp}$ estimation equations to obtain accurate
$OH_{exp}$ in the PAM-OFR has been evaluated by linear regression of $OH_{exp, est}$ against $OH_{exp, dec}$. We have also evaluated how
well estimation equations perform when using limited ranges of experimental parameters (e.g., $OHR_{ext}$, residence time, water
mixing ratio, etc.) or different trace gases ($SO_2$ and CO) and given recommendations. In addition, we have proposed the
minimal number of trace-gas decay experiments required to obtain a set of usable parameters for the $OH_{exp}$ estimation
equations. Finally, we also compared the advantages and disadvantages of the OFR185 and the OFR254 modes from the
perspective of the quantification of $OH_{exp}$. The methodology of this study can be applied to laboratory and field experiments
for $OH_{exp}$ estimation using PAM-OFR or other OFRs that follow a plug-flow assumption.
**2 Methods**
**2.1 The PAM-OFR**
Experiments were conducted using an Aerodyne PAM-OFR (Aerodyne Research Inc., Billerica, MA, US), which is a
horizontal aluminium cylindrical chamber with an internal volume of 13.3 L. The PAM-OFR operates in a continuous flow
mode. Four low-pressure Hg lamps are installed inside the reactor to produce UV light with characteristic spectral lines (e.g.,
185 and 254 nm). The OH is generated via OFR185 using two ozone-producing Hg lamps (GPH436T5VH/4P, Light Sources,
Inc.) or via OFR254 using two ozone-free Hg lamps (GPH436T5L/4P, Light Sources, Inc.) to photolyze externally added
ozone. A flow of nitrogen purge gas, ranging from 0.2 to 0.3 L min$^{-1}$, is introduced between the lamps and sleeves. This
nitrogen gas flow serves to reduce the heat generated by the lamps and prevent the formation and accumulation of ozone
between the lamps and the quartz tubes that isolate them from the sample flow in the OFR. A fluorescent dimming ballast is
used to control the photon flux by regulating the voltage applied to the lamps, which allows us to generate different OH
concentrations. In typical measurement sequences, nine lamp voltage settings (including lights off) were cycled through every
2–3 hours. The dimming voltage ranged from 0 to 10 V direct current (DC).

## 2.2 $OH_{exp}$ estimation through decay of $SO_2$ and CO ($OH_{exp, dec}$)

$OH_{exp}$ can be indirectly measured by detecting the decay of the tracers with known reaction rates. Inorganic trace gases $SO_2$
or CO react with OH radicals at slower rates compared to most VOCs. However, considering the complex oxidation chemistry
of VOCs, $SO_2$ and CO can better capture the features of real $OHR_{ext}$ decay and effective $OHR_{ext}$ (Peng et al., 2015). We
performed systematic decay experiments with $SO_2$ and CO in the PAM-OFR, with conditions tabulated in Tables S1 and S2.
Figure S1 shows the schematics of the experimental setups in the OFR185 and OFR254 modes. In the OFR185 mode, the
injected gas flow at the inlet of the PAM is made up of three sub-flows: (1) The trace-gas flow, i.e. $SO_2$ of 0.2–8.7 ppm or CO
of 10.2–207.5 ppm supplied from gas cylinders (Purity: 99.9 % of $SO_2$, 99.95 % of CO; Shanghai Shenkai Gases Technology
CO., LTD.); (2) dry clean air from a zero-air generator (ZAS-100/150, Convenient) with a non-methane hydrocarbon content
of less than 1 ppb; (3) the humidified clean air passed through a Nafion humidifier (FC100-80-6MSS, Perma Pure). By
adjusting the ratio of dry air to humidified air, the water vapor mixing ratio in the PAM-OFR can be controlled. Additionally,
they also serve as makeup flows to maintain a constant flow rate. At the outlet of the reactor, the gas flow was sampled from
an internal perforated Teflon ring. The gas-phase species ($O_3$, $SO_2$, and CO) were detected using an ultraviolet ozone analyser
(UV-100, Eco Sensors), an $SO_2$ monitor (Model 43i, Thermo Scientific), and a CO monitor (G2401, Picarro), respectively. In
the OFR254 mode, in addition to the previously mentioned setup, externally generated $O_3$ (through UV photolysis) with desired
concentrations was injected at the inlet of the PAM-OFR.
Figures S2a and S2b depict examples of set and measured parameters during experiments conducted in the OFR185 and
OFR254 modes, respectively. In the OFR185 mode, without radical generation to oxidize the tracer species, their concentration
was allowed to stabilize under dark conditions. Once the concentration reached a steady state, the UV lamps were turned on.
Different light intensities lead to varying levels of decay of $SO_2$ or CO after oxidation, reflecting different $OH_{exp}$ within the
PAM-OFR. In the OFR254 mode, it is necessary to obtain the initial concentration of $O_3$ injected into the PAM-OFR in the
absence of $OHR_{ext}$. While waiting for the $SO_2$ or CO concentration to stabilize, the $O_3$ flow was temporarily blocked outside
the PAM-OFR using a valve. Dry clean air was then introduced to compensate for this portion of the flow, ensuring a constant
total flow throughout the entire process. Once the tracer species concentration had reached a steady state, the $O_3$ was then
allowed to flow into the PAM-OFR. The total $OH_{exp, dec}$ in the reactor was varied over a wide range (approximately $10^9$–$10^{12}$
molecules cm$^{-3}$ s) by changing the UV light intensity, water mixing ratio, and residence time. The mean residence time was
obtained from the ratio of the internal volume of and the total flow rate through the PAM-OFR. In the calculation of $OH_{exp, dec}$
(see the paragraph below), plug flow conditions were assumed, which has been shown to agree with the RTD approach for
$OH_{exp}$ when using species (such as $SO_2$ or CO) with low reaction rate constants with OH radicals ($k_{i, OH}$) by Li et al. (2015)
and Peng et al. (2015).
$OH_{exp, dec}$ in the PAM-OFR was calculated from the pseudo-first-order reaction of OH with $SO_2$ or CO, whose $k_{i, OH}$ have been
well characterized ($k_{SO2, OH} = 9.49 \times 10^{-13}$ cm$^3$ molecule$^{-1}$ s$^{-1}$ and $k_{CO, OH} = 2.4 \times 10^{-13}$ cm$^3$ molecule$^{-1}$ s$^{-1}$ at 1 atm and 298 K)
(Burkholder et al., 2020; Cao et al., 2020). By measuring the decay of $SO_2$ or CO, the corresponding $OH_{exp, dec}$ is calculated as
follows:
$$OH_{exp, dec} = \frac{-1}{k_{i, OH}} \times \ln\left(\frac{c_{i, out}}{c_{i, in}}\right) \tag{1}$$
where $c_{i, in}$ is the concentration of reactant $i$ injected into the PAM-OFR (ppb), $c_{i, out}$ is reactant $i$ concentration at the PAM-
OFR outlet (ppb), and $k_{i, OH}$ is the second-order rate constant between the trace species ($SO_2$ or CO) and OH radicals.
Despite the use of nitrogen as a purge gas to reduce the heat generated by the lamp, temperature variations were still observed
within the PAM-OFR. There was a maximum deviation of approximately 13 °C from 25 °C when using $SO_2$ as the OHR
source. However, the $k_{SO2, OH}$ was $8.85 \times 10^{-13}$ cm$^3$ molecule$^{-1}$ s$^{-1}$ at 38 °C (Burkholder et al., 2020), which results in the
calculated $OH_{exp, dec}$ being only approximately 7% higher than that derived from $k_{SO2, OH}$ at 25 °C. Pan et al. (2023) noted that
temperature increases caused by lamp heating exerted minimal influence on gas-phase reaction rates, with $SO_2$ decay and OH
exposure showing negligible variations. Therefore, the influence of temperature on reaction kinetics was not considered in this
study.
**2.3 $OH_{exp, est}$ estimation from empirical equations ($OH_{exp, est}$)**
Li et al. (2015) proposed an $OH_{exp, est}$ estimation equation (Eq. 2) for OFR185 based on easily measurable quantities:
$$OH_{exp, est} = 10^{\left[a+\left(b+c\times OHR_{ext}{}^d+e\times \log\left(O_{3, out}\times\frac{180}{t}\right)\times OHR_{ext}{}^f\right)\times \log\left(O_{3, out}\times\frac{180}{t}\right)+\log H_2O+\log\left(\frac{t}{180}\right)\right]} \tag{2}$$
where $a$–$f$ are fitting parameters (values are reported in Table S5); $O_{3, out}$ is ozone concentration measured at the exit of the
PAM-OFR (molecules cm$^{-3}$), which serves as a surrogate for UV flux; $H_2O$ is water vapor mixing ratio in PAM-OFR (%),
which is influenced by both temperature and relative humidity; t is mean residence time (s). The total external OH reactivity
is represented by $OHR_{ext}$ (s$^{-1}$) $= \sum_i k_i[C_i]$, where $k_i$ and $[C_i]$ are the rate constants with OH and the concentration of the OH-
consuming reactant $i$ in the system (Wang et al., 2020).
Peng et al. (2015) proposed another equation (Eq. (3)) for $OH_{exp, est}$ in OFR254:
$$OH_{exp, est} = 10^{\left[x+\log(-\log r_{O_3})+y\times\left(\frac{OHR_{ext}}{O_{3, in}}\right)^z\right]} \tag{3}$$
where $x$–$z$ are fitting parameters (values are reported in Table S6); log $r_{O3}$ (log ($O_{3, out}/O_{3, in}$)) is the logarithm of the ratio
between the output and input $O_3$ concentrations, which serves as a surrogate for UV flux and also captures the effect of $H_2O$;
$O_{3, in}$ is the concentration of externally injected $O_3$ into the PAM-OFR (molecules cm$^{-3}$).
We have performed in total of 62 sets of trace-gas decay experiments with 416 data points for the $OH_{exp, dec}$, with 25 sets and
175 data points in the OFR185 mode and 37 sets and 241 data points in the OFR254 mode. In OFR185 mode, the 175
experiments cover an $OH_{exp, dec}$ range of $3.57 \times 10^8$–$5.52 \times 10^{12}$ molecules cm$^{-3}$ s, with an equivalent photochemical age ranging
from 4 minutes to 43 days. In OFR254 mode, the 241 experiments cover an $OH_{exp, dec}$ range of $1.01 \times 10^9$–$2.18 \times 10^{12}$ molecules
cm$^{-3}$ s, with an equivalent photochemical age ranging from 11 minutes to 17 days. The error in $OH_{exp, dec}$ is derived from the
measurement error of the tracer gas, propagated through Eq. 1. When $OH_{exp, dec}$ ranged from $3.6 \times 10^8$–$5.5 \times 10^{12}$ molecules
cm$^{-3}$ s, the resulting error values were $1.9 \times 10^8$–$2.4 \times 10^{10}$ molecules cm$^{-3}$ s.
After obtaining the $OH_{exp, dec}$ values, we used Eqs. 2 and 3 to fit the parameters $a$–$f$ and $x$–$z$ for OFR185 and OFR254 modes,
respectively, given that the experimental parameters such as $OHR_{ext}$, $O_{3, out}$, $H_2O$, and t (in Eq. 2), and $r_{O3}$, $OHR_{ext}$, and $O_{3, in}$
(in Eq. 3) are known. The $OH_{exp, est}$ values were then reconstructed with the fitted parameters and the experimental parameters,
and compared with the $OH_{exp, dec}$ values via linear regression analysis. Similarly, the error values for all $OH_{exp, est}$ values are at
least one order of magnitude smaller than the respective $OH_{exp, est}$ values. The generation of OH radicals in PAM-OFR is related
to the photon fluxes at $\lambda = 185$ nm ($I_{185}$) and $\lambda = 254$ nm ($I_{254}$). According to Rowe et al. (2020), $I_{185}:I_{254}$ is specific to the Hg
lamp utilized. Since the $OH_{exp}$ estimation equation for OFR185 uses $O_3$ concentration as a measurable surrogate for the UV
flux at 185 nm, it is also lamp-specific. Because the UV lamps used in our study are different from the BHK lamps employed
by Li et al. (2015), we anticipate that the parameters $a$–$f$ fitted from our decay experiments (Table S5) should be quite different
from those in Li et al. (2015), which is indeed the case. Similarly, fitting parameters $x$–$z$ for OFR254 mode from our decay
experiments (Table S6) are also different from those in Peng et al. (2015).
**3 Results and Discussion**
**3.1 The OFR185 mode: $OHR_{ext}$ level relevant to ambient conditions**
Field studies showed that the environmental $OHR_{ext}$ mainly fluctuated between 10–30 s$^{-1}$ (Fuchs et al., 2017; Lou et al., 2010;
Lu et al., 2010; Tan et al., 2018; Yang et al., 2017). To investigate the factors that potentially affect the fitting parameters of
Eq. 2 in the estimation of $OH_{exp}$ under ambient conditions, we first performed 16 sets of experiments with $OHR_{ext}$ of 4–23 s$^{-1}$
using $SO_2$ as the $OHR_{ext}$ source. With the measured $OH_{exp, dec}$, the parameters ($a$–$f$) were first derived, which were used to
reconstruct $OH_{exp, est}$ using Eq. 2 with known $OHR_{ext}$, ozone concentration ($O_{3, out}$), water vapor mixing ratio ($H_2O$), and
residence time (t). The reconstructed $OH_{exp, est}$ values were plotted against the $OH_{exp, dec}$ values calculated from the trace-gas
decay experiments, as shown in Figure 1. The 1:2 and 2:1 lines indicate approximately half an order of magnitude difference
between $OH_{exp, dec}$ and $OH_{exp, est}$, which is considered to be acceptable as an uncertainty in $OH_{exp}$ estimation.
We first investigated the effect of changing residence time on the $OH_{exp}$ estimation. With other experimental parameters (i.e.
$H_2O$, $O_{3, out}$, and $OHR_{ext}$) being similar, we set the residence time to a low value (33 s) and also a range of higher values (61–
200 s). The detailed ranges of each experimental condition for different datasets are listed in Table S3. With the residence time
of 33 s, the reconstructed $OH_{exp, est}$ correlates well with the experimental $OH_{exp, dec}$ (slope = 1.061 and $R^2$ = 0.990, Figure 1a1).
The set of fitted parameters $a$–$f$ ($FP_{st, 185}$; st: short time) applied in Figure 1a1 is presented in Table S5. When the residence
time was increased to 61–200 s, the interpolated $OH_{exp, est}$ utilizing $FP_{st, 185}$ was also in good correlation with $OH_{exp, dec}$ (slope
= 0.978, $R^2$ = 0.959, Figure 1a2).  We also derived fitted parameters ($FP_{et, 185}$; et: extended t) using the data points with the
extended range of residence time (33–200 s). Not surprisingly, with the application of $FP_{et, 185}$, $OH_{exp, est}$ also correlated well
with $OH_{exp, dec}$ (slope = 0.994, $R^2$ = 0.955, Figure 1a3). The results indicate that variation in residence time does not significantly
affect the fitting parameters of Eq. 2 for the $OH_{exp}$ estimation. From an experimental perspective, since $OH_{exp}$ is the product
of OH radical concentration ([OH]) and the residence time (t), as long as the change of t does not significantly alter the quasi-
steady-state [OH], the fitted parameters from a narrow range of t should be applicable to situations of longer t. Mathematically,
two terms of 180/t and t/180 are related to t, ranging from 0.90–5.45 and 0.18 to 1.11, respectively, which do not contribute
significantly to the exponent in Eq. 2 after taking the logarithm of them. It is important to note that the above discussion
regarding residence time assumes a plug-flow condition within the PAM-OFR, which is applicable to substances with low $k_{i, OH}$,
such as $SO_2$ (or CO). For species that react rapidly with OH, such as monoterpenes or toluene, localized concentration
gradients can develop within the OFR, leading to a significant uneven actual RTD that affects the estimation of $OH_{exp}$ (Palm
et al., 2018).
Similarly, we then investigated the impacts of $H_2O$ on the estimation of $OH_{exp}$. Applying fitted parameters from experiments
of low water vapor mixing ratios (0.49–0.99 %, Figure 1b1) ($FP_{lH2O, 185}$; $lH_2O$: low $H_2O$) to data spanning a wide range of
water vapor mixing ratios (0.49–2.76 %) also yielded a reasonably good correlation between $OH_{exp, est}$ and $OH_{exp, dec}$ (Figure
1b2). This could be attributed to the fact that the term $logH_2O$ in Eq. 2 does not contribute significantly to the exponent.
As for ozone concentration, applying fitting parameters ($FP_{lO3, 185}$; $lO_3$: low $O_{3, out}$) from experiments of low ozone
concentration ($1.44 \times 10^{12}$–$6.79 \times 10^{13}$ molecules cm$^{-3}$, Figure 1c1) to reconstruct the data for a wide range ($1.44 \times 10^{12}$–2.03
$\times 10^{15}$ molecules cm$^{-3}$) yielded a reasonably good correlation between $OH_{exp, est}$ and $OH_{exp, dec}$ (Figure 1c2). It only resulted in
a mildly increased slope (from 1.063 to 1.272) and similar $R^2$ values (both are 0.970) as compared to those using the whole
ozone concentration range (Figure 1c3).
Ideally, trace-gas decay experiments covering the entire ranges of the t, $H_2O$, and $O_{3, out}$ variations under real experimental
conditions should be conducted, which is labor-intensive. Practically, due to the atmospherically relevant variations that occur
in t, $H_2O$, and $O_{3, out}$ during the real experiments, the ranges of t, $H_2O$, and $O_{3, out}$ covered by trace-gas decay experiments are
usually narrower compared to the real experiments. Our results suggest that the fitting parameters ($a$–$f$) obtained from
calibration experiments with relatively narrow ranges of t, $H_2O$, and $O_{3, out}$ can still provide a reliable estimation of OH radical
levels during the real experiments, which would cover wider ranges of these conditions.
It is noteworthy that reliable estimations can be achieved regardless of whether the narrow range is situated within the lower
or higher interval of the full condition range. Figure 1 demonstrated the case where the narrow range was situated within the
lower interval, while Figure S3 presented the case where the narrow range was situated within the higher interval. The detailed
ranges of each experimental condition for different datasets are listed in Table S3. As shown in Figure S3, the data points in
panel a1 had residence times of 100–296 s, the data points in panel b1 had water vapor mixing ratios of 1.04–2.76 %, and the
data points in panel c1 had $O_{3,\,out}$ of $8.45 \times 10^{13}$–$2.03 \times 10^{15}$ molecules $cm^{-3}$. Panels a2, b2, and c2 built on panels a1, b1, and
c1 by incorporating data points with shorter t (33–61 s), lower $H_2O$ (0.49–0.97 %), and lower $O_{3,\,out}$ ($1.44 \times 10^{12}$–$6.79 \times 10^{13}$
molecules $cm^{-3}$), respectively, but still used fitting parameters *a–f* obtained from the higher range of conditions to estimate
$OH_{exp,\,est}$. In panels a3, b3, and c3, the parameters *a–f* were refitted using all the data points included in the expanded t, $H_2O$,
and $O_{3,\,out}$ ranges, respectively, and the obtained *a–f* were used to estimate $OH_{exp,\,est}$. Using panel a1–a3 in Figure S3 as an
example, the slope and $R^2$ values in a2 and a3 were very close to 1, reflecting the good consistency between $OH_{exp,\,est}$ and
$OH_{exp,\,dec}$. In the OFR254 mode discussed later (Figure 4, panels c1–c3), this narrower range can also be situated within the
middle interval of the full condition range. This applicability of fitting parameters obtained from narrow ranges of experimental
conditions is beneficial for quickly obtaining concurrent $OH_{exp}$ during the experiments in field measurements.

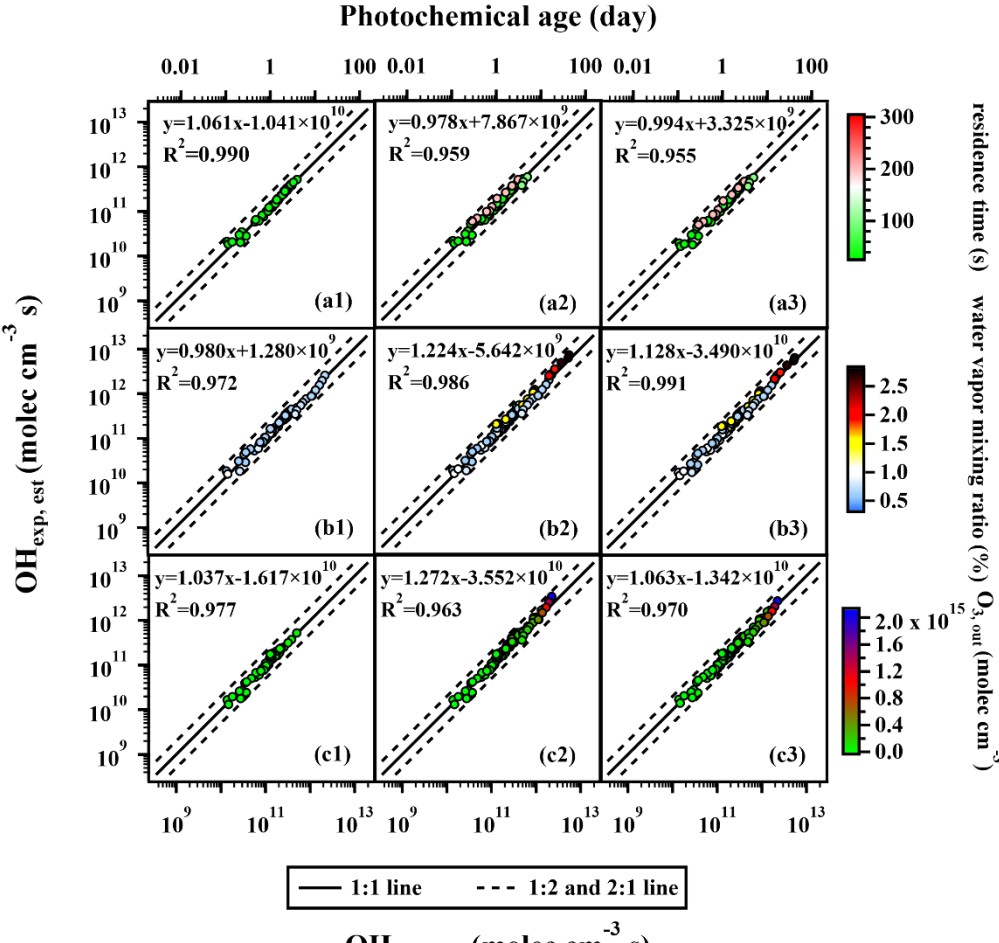

**Figure 1: The regression results of $OH_{exp, est}$ and $OH_{exp, dec}$ when variations occurred in (a1–a3) residence time, (b1–b3) water vapor mixing ratio, and (c1–c3) output $O_3$ concentration under atmospheric relevant $OHR_{ext}$ level (4–23 $s^{-1}$). Compared to panels a1, b1, and c1, panels a2, b2, and c2 respectively incorporated additional data points with higher t, $H_2O$, and $O_{3, out}$ values, but still utilized the fitting parameters $FP_{st, 185}$, $FP_{lH2O, 185}$, and $FP_{lO3, 185}$ obtained from the lower condition range to estimate $OH_{exp, est}$. In panels a3, b3, and c3, all data points within the extended condition range were used to re-fit the parameters a–f, and the resulting $FP_{et, 185}$, $FP_{eH2O, 185}$, and $FP_{eO3, 185}$ were employed to estimate $OH_{exp, est}$ (s: short, l: low, e: extended). All the error values for $OH_{exp, dec}$ are half or even two orders of magnitude smaller than the corresponding $OH_{exp, dec}$ values. When applying a logarithmic scale, the error bars become difficult to represent. To enhance the readability of the graph, error bars have not been included. For the same reason, error bars for $OH_{exp, est}$ values are also not displayed.**

### 3.2 The OFR185 mode: $OHR_{ext}$ level relevant to emission sources

The experimental conditions in the PAM-OFR often involve not only general atmospheric conditions ($OHR_{ext} < 30$ $s^{-1}$) but also high-concentration conditions, e.g., those directly from emission sources. For instance, the $OHR_{ext}$ of direct vehicle emission can be as high as 1000 $s^{-1}$ with plenty of reducing gases such as CO and VOCs (Nakashima et al., 2010). To evaluate the applicability of Eq. 2 under situations of high $OHR_{ext}$, we performed high $OHR_{ext}$ (up to 204 $s^{-1}$) experiments using high

concentrations of $SO_2$ as the $OHR_{ext}$ source. Compared to the data points shown in Figure 2a (4–23 s$^{-1}$), Figure 2b and Figure 2c included additional data points with higher $OHR_{ext}$ values (198–204 s$^{-1}$), while the other conditions remained similar. In Figure 2b, the parameters $a$–$f$ (FP$_{lOHR, 185}$; lOHR: low $OHR_{ext}$) obtained from the low-$OHR_{ext}$ data points were used to estimate $OH_{exp, est}$, yet those used in Figure 2c were refitted from the data points with extended $OHR_{ext}$ range (4–204 s$^{-1}$). It could be observed from Figure 2b that when estimating $OH_{exp}$ using FP$_{lOHR, 185}$, $OH_{exp, est}$ of the high-$OHR_{ext}$ data points were significantly overestimated, with a difference of more than two orders of magnitudes compared to $OH_{exp, dec}$. This observation suggests that, different from cases for residence time, water vapor mixing ratio, and ozone concentration shown in the section above, FP$_{lOHR, 185}$ were not applicable to high-$OHR_{ext}$ conditions.

We then investigated the possible causes of the discrepancy for $OH_{exp}$ estimation between FP$_{lOHR, 185}$ and FP$_{eOHR, 185}$ (eOHR: extended $OHR_{ext}$). From a mathematical perspective, according to Eq. 2, the third term $c \times OHR_{ext}^d \times \log(O_{3, out} \times 180/t)$ and the fourth term $e \times OHR_{ext}^f \times [\log(O_{3, out} \times 180/t)]^2$ are associated with $OHR_{ext}$, which involve fitted parameters of $c$–$f$. To investigate their relationships with $OHR_{ext}$, we performed a sensitivity test with a fixed ozone concentration ($1.77 \times 10^{14}$ molecules cm$^{-3}$) and residence time (89 s), which were mean values during our experiments. When using the $c$–$f$ values of FP$_{lOHR, 185}$ (-0.13922, 0.26786, 0.0026332, and 0.4917), the variations of the third term, the fourth term, and their sum with respect to $OHR_{ext}$ were shown in Figure S4a1–a3, respectively. The third term (Figure S4a1) was negative and decreased as $OHR_{ext}$ increased, while the fourth term (Figure S4a2) was positive and increased as $OHR_{ext}$ increased. The sum of them (Figure S4a3), however, first decreased and then started to increase at approximately $OHR_{ext} = 21$ s$^{-1}$, owing possibly to a slower decrease in the third term or a faster increase in the fourth. If contributions from other terms in Eq. 2 were constant, this led to an increase of $OH_{exp}$ as $OHR_{ext}$ increased beyond 21 s$^{-1}$. Our results showed that the expectation that $OH_{exp}$ should decrease with increasing $OHR_{ext}$ (Li et al., 2015) was applicable to the lower ranges of $OHR_{ext}$, i.e., under atmospheric relevant conditions. With further increase of $OHR_{ext}$, i.e., above atmospheric relevant condition, the fitted parameters obtained from the dataset with FP$_{lOHR, 185}$ were not applicable.

When using the $c$–$f$ values of FP$_{eOHR, 185}$ (-0.079114, 0.36805, 0.0041654, and 0.38722), the trends of the third and the fourth terms (Figure S4b1 and S4b2, respectively) were similar to those with low $OHR_{ext}$ (Figure S4a1 and S4a2, respectively); their sum, however, gave a monotonical decreasing trend as $OHR_{ext}$ increased (Figure S4b3), consistent with the expectation that $OH_{exp}$ should decrease with increasing $OHR_{ext}$ (Li et al., 2015). The curve in Figure S4b3 can continue to decrease monotonically at higher $OHR_{ext}$ values, at least until 2000 s$^{-1}$.

From the perspective of oxidation chemistry, high concentrations of gas phase $SO_2$ could lead to more $SO_2$ entering the particle phase. The $H_2O_2$ in the liquid water of nucleated sulfuric acid aerosols would further oxidize $SO_2$ (Liu et al., 2020), which could lead to the discrepancy for OH estimation between low OHR and extended high OHR.

Nevertheless, the good agreement between $OH_{exp, est}$ and $OH_{exp, dec}$ in Figure 2c (using re-fitted parameters from the dataset of extended $OHR_{ext}$) indicate that Eq. 2 can still be used to estimate $OH_{exp}$ under high-$OHR_{ext}$ conditions. This conclusion is further supported by the results of $OH_{exp}$ obtained using CO as the $OHR_{ext}$ source (see Figure 3 and the section below) under extremely high-$OHR_{ext}$ conditions (up to 1200 s$^{-1}$). This is advantageous for the use of PAM-OFR in simulations of SOA

formation from direct emission sources (e.g., vehicular exhaust and biomass burning) where $OHR_{ext}$ is extremely high. It is,
however, desirable to have $OH_{exp}$ estimated under similarly high $OHR_{ext}$ for those experiments to accurately represent the
extent of oxidation.

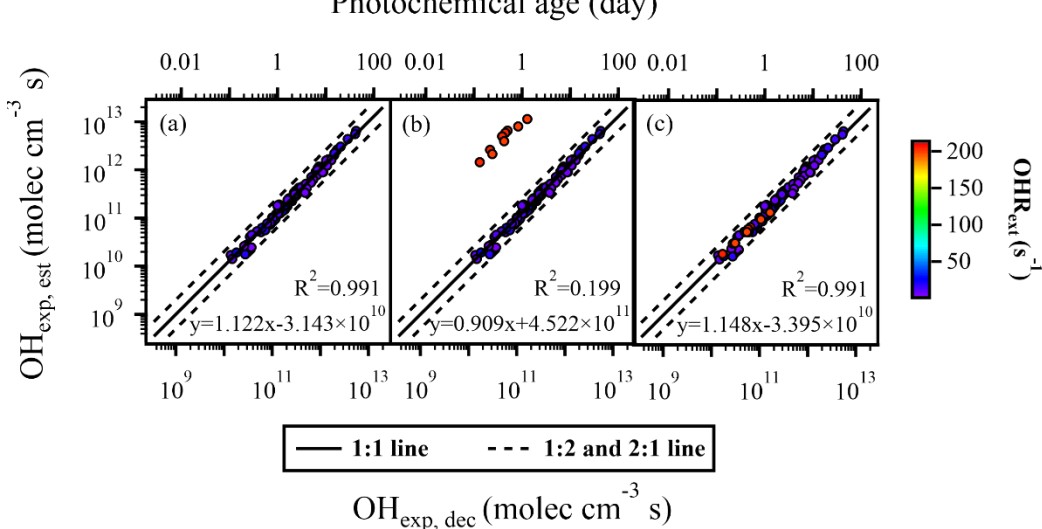


**Figure 2: The regression results of $OH_{exp, est}$ and $OH_{exp, dec}$ with different $OHR_{ext}$ levels. In panel a, data points with atmospheric**
**relevant $OHR_{ext}$ level (4–22 $s^{-1}$) were applied. In addition to the data points contained within panel a, panel b included additional**
**data points with emission sources related $OHR_{ext}$ level (198–204 $s^{-1}$), but $FP_{lOHR, 185}$ were still used to estimate $OH_{exp, est}$. In panel b,**
**data points in red showed that the $OH_{exp, est}$ of these high-$OHR_{ext}$ data points were significantly overestimated. $FP_{lOHR, 185}$ were not**
**applicable to high-$OHR_{ext}$ conditions. The data points in panel c were identical to those in panel b, but the estimation of $OH_{exp, est}$**
**utilized the $FP_{eOHR, 185}$ obtained by fitting all data points across the full range of $OHR_{ext}$ levels.**
**3.3 The OFR185 mode: $SO_2$ and CO as $OHR_{ext}$ sources**
Peng et al. (2015) suggested that $SO_2$ can better capture the features of real $OHR_{ext}$ decay and effective $OHR_{ext}$. The reaction
between $SO_2$ and OH is relatively straightforward and is not expected to undergo too many side reactions. CO is a typical
gaseous inorganic compound emitted during combustion process. Using CO as an $OHR_{ext}$ source to explore the estimation of
$OH_{exp}$ in the simulation of oxidation chemistry for emission sources (i.e., high $OHR_{ext}$ level) is representative. Therefore, we
compared the results with $SO_2$ (Figure 3a) and CO (Figure 3b) as the $OHR_{ext}$ source. When using $SO_2$ as the $OHR_{ext}$ source,
all data points agreed within a factor of 2 (Figure 3a). while only approximately 83 % of the data points agreed within a factor
of 2 when CO was used as the $OHR_{ext}$ source (Figure 3b). The deviating data points were mostly concentrated in areas with
high $OHR_{ext}$ (> 600 $s^{-1}$) and low $O_{3, out}$ concentration ($10^{12}$–$10^{13}$ molecules $cm^{-3}$), where the removal of CO was relatively low.
Li et al. (2015) have observed increased deviations between $OH_{exp, est}$ and $OH_{exp, dec}$, which was attributed, at least in part, to
the increased measurement uncertainties for CO when the decrease of its concentration was marginal. We believe that
measurement uncertainty might not be the main reason in our case, because most of the decreases in CO concentration during
our experiments were larger than the precision of the Picarro G2401 Analyzer (~1.5 ppb at 5 min time resolution). Another
possible reason is that in addition to the reaction with OH radicals, CO may react with some other oxidants, leading to its
consumption, while $SO_2$ was less affected, thereby resulting in more scattered data points for CO. The reaction rate of CO with
$HO_2$ is very slow, and is unlikely to play a significant role ($k_{CO, HO2} = 5.55 \times 10^{-27}$ cm$^3$ molecule$^{-1}$ s$^{-1}$ at 300 K) (You et al.,
2007). Cohen and Heicklen (1972) suggested that CO could also react with atomic oxygen (O($^1$D)). Clerc and Barat (1967)
have reported some appreciable rate coefficients ($10^{-11}$ to $10^{-12}$ cm$^3$ molecule$^{-1}$ s$^{-1}$) for the reaction between CO and O($^1$D),
which are higher than those for the reactions of CO with OH ($k_{CO, OH} = 2.4 \times 10^{-13}$ cm$^3$ molecule$^{-1}$ s$^{-1}$ at 298 K) (Burkholder et
al., 2020). It is therefore possible that reaction between CO and O($^1$D) might have complicated the decay of CO in the PAM-
OFR. To further investigate this aspect, we used the KinSim, a kinetic simulator, to calculate the average mixing ratios of OH,
O($^1$D), and $HO_2$ under the specific conditions in the PAM-OFR, and then assessed the relative importance of the reactions CO
+ OH → $CO_2$ + H, CO + O($^1$D) → $CO_2$, and CO + $HO_2$ → $CO_2$ + OH (Li et al., 2015; Peng and Jimenez, 2019, 2020). The
results show that although the reaction rate constant of CO and O($^1$D) is 1–2 orders of magnitude higher than that of CO and
OH, the concentration of OH is about 6–7 orders of magnitude higher than the concentration of O($^1$D), indicating that the
reaction of CO with O($^1$D) will not have a significant impact on the consumption of CO. The real reason for the scattered data
points when using CO in the trace-gas decay experiment is still unknown.
Figure 3c includes the results of trace-gas decay experiments using both $SO_2$ and CO as the $OHR_{ext}$ source. Despite having
different reaction rates with OH radicals, the data points could be collectively utilized to fit the parameters for the estimation
equation. With approximately 95 % of the results agreeing within a factor of 2, $OH_{exp, est}$ obtained using the fitted parameters
exhibited good agreements (slope = 1.101, $R^2$ = 0.991) with $OH_{exp, dec}$. Our results thus suggest that although using CO as the
$OHR_{ext}$ might result in some scattered data points, it was still feasible to use Eq. 2 to estimate $OH_{exp}$ given that experiments
were not done solely in conditions with high $OHR_{ext}$ (i.e., high CO concentrations) and low $O_3$ concentrations. Another benefit
of using CO as $OHR_{ext}$ source for the estimation of $OH_{exp}$ is that it introduces complexity in the precursor, which resembled
those in real applications. Although not tested in this study, we also note that further trace-gas decay experiments in the
presence of $N_2O/NO_x$ (typical urban environment) should be conducted when oxidation chemistry in the presence of $NO_x$ is
studied (Cheng et al., 2021).

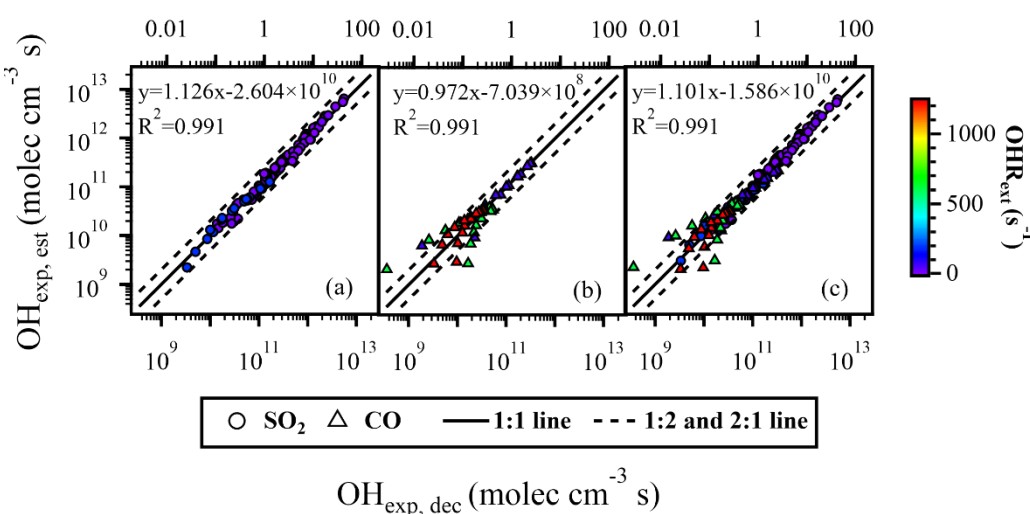


**Figure 3: The regression results of $OH_{exp, dec}$ and $OH_{exp, est}$ in the OFR185 mode with (a) $SO_2$ and (b) CO as $OHR_{ext}$ sources. (c) Results from all experiments (using $SO_2$ and CO) in the OFR185 mode.**

**3.4 The OFR254 mode**
The equation for $OH_{exp}$ estimation in OFR254 mode is simpler compared to that of OFR185 mode. According to Eq. 3, under
OFR254 mode, the three parameters potentially affecting the $OH_{exp}$ are $OHR_{ext}$, input $O_3$ concentration, and $r_{O3}$. The detailed
ranges of each experimental condition for different datasets are listed in Table S4. We found that compared to Figure 1, the
data points in Figure 4 were more scattered. Most of the $R^2$ values in Figure 4 were below 0.9, indicating that using $SO_2$ as the
$OHR_{ext}$ source, the estimation of $OH_{exp}$ (using Eq. 3) under the OFR254 mode performed not as well as those under the OFR185
mode (using Eq. 2). Firstly, we investigated the impacts of $OHR_{ext}$. Figure 4a1 showed the regression results of $OH_{exp, est}$ and
$OH_{exp, dec}$ when $OHR_{ext}$ ranged from 5 to 14 $s^{-1}$. The parameters $x$–$z$ ($FP_{lOHR, 254}$; lOHR: low external OHR) (Table S6) were
obtained by fitting Eq. 3 to $OH_{exp, dec}$. In Figure 4a2, the same set of fitted parameters $FP_{lOHR, 254}$ from Figure 4a1 were used for
a wider range of $OHR_{ext}$ (5–21 $s^{-1}$). From the regression results (slopes of 1.050 and 1.024, $R^2$ of 0.890 and 0.894), the same
set of parameters yielded similar estimation performance for $OH_{exp}$ despite a wider range of $OHR_{ext}$ in Figure 4a2 compared
to that of Figure 4a1. At the same time, these results were not much different from those (slope = 1.071, $R^2$ = 0.891) using a
re-fitted set of parameters ($FP_{eOHR, 254}$; eOHR: extended external OHR) for the wider range of $OHR_{ext}$ (Figure 4a3). Even
though the correlation was not as good as those in the OFR185 mode, approximately 85 % of the data points agreed within a
factor of 2. We did not further extend the $OHR_{ext}$ to values as high as those in the OFR185 mode as discussed above, since the
OFR254 mode was much less oxidative and might not be suitable for simulating the oxidation chemistry of extremely high
$OHR_{ext}$ as those from direct emissions.
Similarly good correlations were observed when we only used the fitted parameters ($FP_{lO3, 254}$ and $FP_{mrO3, 254}$, respectively; $lO_3$:
low $O_{3, in}$, $mrO_3$: medium $r_{O3}$) from narrow ranges of input $O_3$ concentration and $r_{O3}$ (Figure 4b1 and Figure 4c1, respectively)
to reconstruct the $OH_{exp, est}$ values with extended ranges of these experimental conditions (Figure 4b2 and Figure 4c2,
respectively). Such correlations were as good as those with re-fitted parameters ($FP_{eO3, 254}$ and $FP_{erO3, 254}$, respectively; $eO_3$:
extended $O_{3, in}$, $erO_3$: extended $r_{O3}$) from data points in the extended ranges of $O_3$ concentration and $r_{O3}$ (Figure 4b3 and Figure
4c3, respectively). These observations thus indicate that under the OFR254 mode, when $OHR_{ext}$, $O_{3, in}$, and $r_{O3}$ vary within
certain ranges (5–21 $s^{-1}$, $6.46 \times 10^{13}$–$4.8 \times 10^{14}$ molecules $cm^{-3}$, and 0.61–0.99, respectively), Eq. 3 can be used to estimate
OH radical levels reasonably well using the fitting parameters ($x$–$z$) obtained from a narrower range of data points.

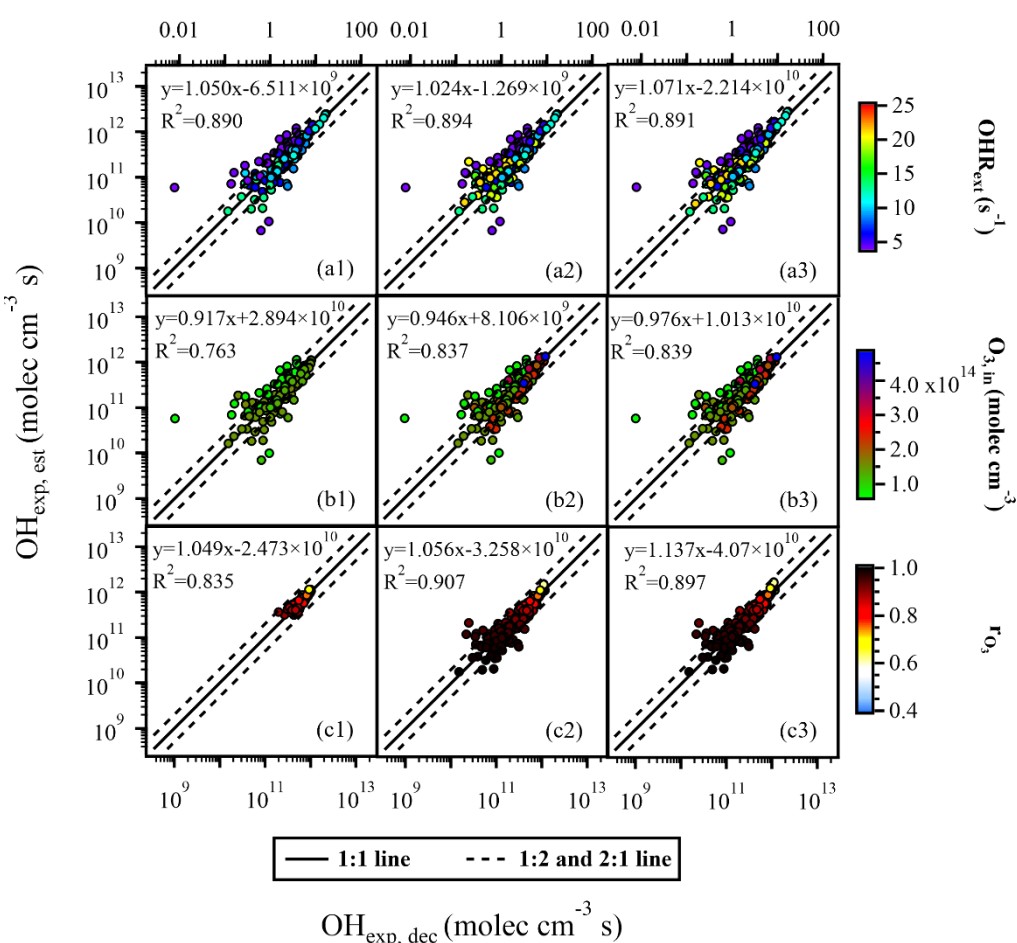


**Figure 4: The regression results of $OH_{exp, est}$ and $OH_{exp, dec}$ when variations occurred in (a1–a3) $OHR_{ext}$, (b1–b3) input $O_3$**
**concentration, and (c1–c3) $r_{O3}$. Compared to panels a1, b1, and c1, panels a2, b2, and c2 respectively incorporated additional data**
**points with extended $OHR_{ext}$, $O_{3, in}$, and $r_{O3}$ values, but still utilized the fitting parameters $FP_{lOHR, 254}$, $FP_{lO3, 254}$, and $FP_{mrO3, 254}$**
**obtained from the lower or medium condition range to estimate $OH_{exp, est}$. In panels a3, b3, and c3, all data points within the extended**
**condition range were used to re-fit the parameters *x–z*, and the resulting FP_eOHR, 254, FP_eO3, 254, and FP_erO3, 254 were employed to**
**estimate OH_exp, est.**
Figure 5a and Figure 5b depicted the correlation between $OH_{exp, est}$ estimated from Eq. 3 and $OH_{exp, dec}$ calculated from Eq. 1
with $SO_2$ and CO as $OHR_{ext}$ sources, respectively. When using $SO_2$ as the $OHR_{ext}$ source, approximately 86 % of the data
points agreed within a factor of 2 (Figure 5a). Similar to the case of OFR185, when CO was used as the $OHR_{ext}$ source, the
data points were more scattered, with the percentage of data points within a factor of 2 dropping to only about 64 % (Figure
5b). Figure 5c included data points using both $SO_2$ and CO as the $OHR_{ext}$ sources. Overall, regardless of the $OHR_{ext}$ source,
when $r_{O3}$ was higher than 0.93, which meant a low UV intensity, the majority of data points for $OH_{exp, est}$ and $OH_{exp, dec}$ differed
by a factor of two or more. It is therefore recommended that when using the OFR254 mode, too low lamp power settings, for
example, UV lamp voltage below 1.5 V should be avoided in the case of our study.

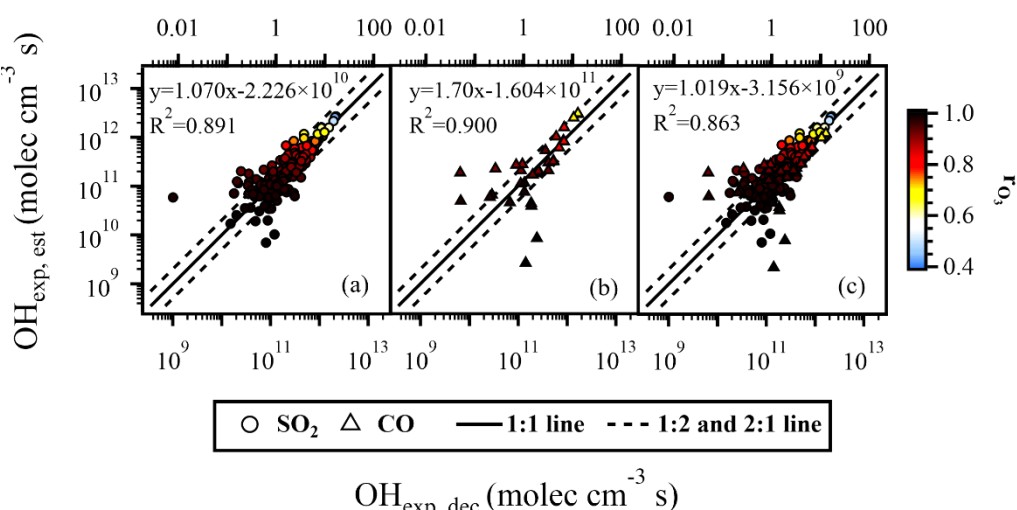


**Figure 5: The regression results of OH_exp, dec and OH_exp, est in the OFR254 mode with (a) SO₂ and (b) CO as OHR_ext sources. (c)**
**Results from all experiments (SO₂ and CO) in the OFR254 mode.**

## 4 Conclusions

A series of $OH_{exp}$ estimation experiments using the PAM-OFR were conducted in OFR185 and OFR254 modes to explore the
applicability of the empirical equations under a wide range of conditions. The results indicate that for OFR185 mode, when
varying the residence time, water vapor mixing ratio, and output $O_3$ concentration (as a surrogate for UV intensity) within
certain ranges, the empirical equation (Eq. 2) for $OH_{exp}$ proves to be effective in estimating $OH_{exp}$. Unless there is a significant
change in $OHR_{ext}$, such as transitioning from ambient conditions to emission source conditions, there is no need to re-fit the
parameters *a–f* in the estimation equation to estimate $OH_{exp}$. Compared with OFR254 mode, the consistency between $OH_{exp,}$
_est_ and $OH_{exp, dec}$ in OFR185 mode is better. For the OFR254 mode, when $OHR_{ext}$, input $O_3$ concentration, and $r_{O3}$ vary within
certain ranges, the empirical equation (Eq. 3) can be used to estimate $OH_{exp}$ reasonably well using the parameters $x$–$z$ obtained
from a narrower range of data points. It is important to note that for the OFR185 mode, the above conclusions are valid only
if one already has a set of $a$–$f$ values that are appropriate for the specific UV lamps being used, as the $I_{185}$:$I_{254}$ that affects the
$OH_{exp}$ is lamp-specific. For a PAM-OFR that employs a different Hg lamp, a series of calibration experiments should be
conducted in any case. Alternatively, based on the research by Rowe et al. (2020), the exponential relationship between the $a$–
$f$ values and the $I_{185}$:$I_{254}$ could be used to first obtain a set of $a$–$f$ values suitable for the UV lamps being used.
To obtain reliable estimates of $OH_{exp}$ using Eqs. 2 and 3 for the OFR185 mode or OFR254 mode, respectively, it is desirable
to have sufficient data points (that is, $OH_{exp, dec}$ from trace-gas decay experiments) to fit the parameters for the calculation of
$OH_{exp, est}$. There is currently no consensus on how many data points in trace-gas decay experiments are enough for reliable
fitted parameters, which could be important for in-situ $OH_{exp}$ estimation in field studies where a limited number of experiments
are done to reduce downtime. We aim to address this by random sampling from the data points in our experiments and
determine the minimum number of experiments that are needed to obtain reliable $OH_{exp}$.
For OFR185 mode, we first used randomly selected N data points from the 175 data points presented previously to fit the
parameters ($a$–$f$) using Eq. 2. The fitted parameters were then used to reconstruct $OH_{exp, est}$ for all the 175 data points. The
$OH_{exp, est}$ values were then compared with the corresponding 175 $OH_{exp, dec}$ values. This procedure was repeated 10 times for
each N, with N starting from 7 till approximately 50 (Figure 6a). The average $R^2$, slope, and intercept from the 10 attempts
were then shown as a function of N for experiments with $SO_2$ only (Figure 6a) and those with $SO_2$ and CO (Figure 6b). It can
be observed that around 30 data points are needed for experiments with $SO_2$ only while around 20 data points are needed to
have stable $R^2$ values and slopes when using both $SO_2$ and CO. For OFR254 mode, the same procedure was applied to the 241
data points. It was not surprising that the results were a lot more scattered (Figure 6c and Figure 6d) compared to those for
OFR185 mode given their performance shown in the previous section. Nevertheless, our analysis suggests that around 25 data
points are needed to obtain reliable $OH_{exp, est}$ for OFR254 mode, whether $SO_2$ alone (Figure 6c) or $SO_2$ and CO (Figure 6d) are
used for the trace-gas decay experiments. Therefore, despite the limitation that this practice only randomly samples the data
points without considering the range of any experimental conditions, our analysis suggests that 20–30 data points are normally
needed to obtain reliable $OH_{exp}$ for both OFR185 and OFR254 modes.

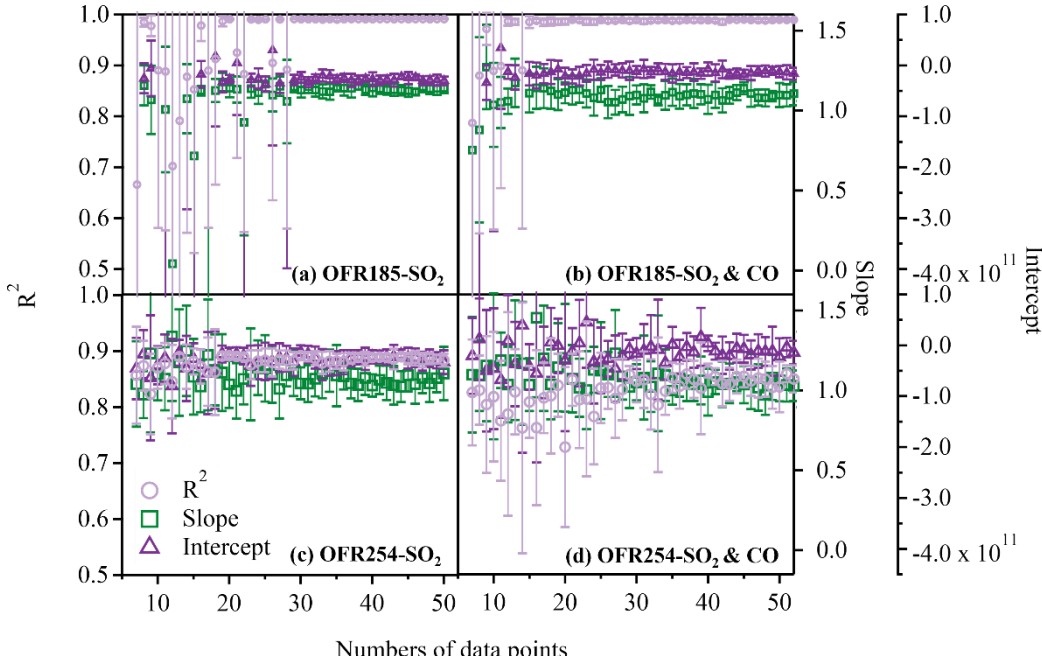


**Figure 6: The regression results of $OH_{exp, dec}$ and $OH_{exp, est}$ (characterized by the $R^2$, slope, and intercept) when different numbers of data points were chosen. (a) $SO_2$ as $OHR_{ext}$ source in OFR185 mode, (b) $SO_2$ or CO as $OHR_{ext}$ source in OFR185 mode, (c) $SO_2$ as $OHR_{ext}$ source in the OFR254 mode, and (d) $SO_2$ or CO as $OHR_{ext}$ source in the OFR254 mode.**

Our study suggests that the $OH_{exp, est}$ estimated from the empirical equations agrees better with $OH_{exp, dec}$ for the OFR185 (Figure 3) than for the OFR254 mode (Figure 5). This can be understood from the perspective of OH generation and its consumption by $OHR_{ext}$ (Li et al., 2015). For the OFR185 mode, there are two pathways to generate OH radicals: the photolysis of $H_2O$ and the photolysis of $O_3$. For the OFR254 mode, the main pathway for OH radical generation is solely the photolysis of $O_3$. Consequently, when $OHR_{ext}$ changes, the disruption to $OH_{exp}$ in the system is more significant in the case of the OFR254 mode, while the $OH_{exp}$ in the OFR185 mode remains more stable. In addition, pseudo-first-order kinetics between OH radicals and $SO_2$ or CO is assumed, with [OH] being at a pseudo-steady state. Yet, the relatively low OH radical generation capacity in the OFR254 mode might not necessarily always fulfil such an assumption, leading to higher uncertainties for estimating $OH_{exp}$. Therefore, the OFR185 mode offers certain advantages such as relatively high $OH_{exp}$, more accurate $OH_{exp}$ estimation, as well as no external input of $O_3$ needed. However, for substances that exhibit strong absorption at the wavelength of 185 nm and are prone to photolysis, such as aromatic species (Peng et al., 2016), using the OFR254 mode is a better choice. For users of other OFRs (non-PAM-OFR) who would like to apply the conclusions above, at least two conditions must be met: (1) the concentration of [OH] within the OFR should remain stable, and (2) the assumption of plug flow conditions is acceptable, allowing for the neglect of differences in the actual RTD, heterogeneity in the UV light intensity and the concentration of

radicals/oxidants at different points within the reactor, which are caused by different designs of reactors (such as wall materials,
shapes, or volumes).

**Data availability**

The data shown in the paper are available on request from the corresponding authors (huangdd@saes.sh.cn and
yongjieli@um.edu.mo).

**Supplement link**

The supplement related to this article is available online at:

**Author contribution**

QL, DDH, and YJL conceived and planned the experiments. QL and YW carried out the experiments. QL, DDH, and YJL
analysed the data and took the lead in writing the paper. QL, DDH, YJL, ATL, and XC contributed to the interpretation of the
results. SL, LZ, CYH, ST, QC, KIH, HW, KMM, and CH provided significant input during the revision of the manuscript. All
authors provided feedback on the paper.

**Competing interests**

The authors declare no competing interests.

**Acknowledgements**

Dan Dan Huang acknowledges the financial support from the National Key Research and Development Program of China
(2022YFC3703600), the Science and Technology Commission of Shanghai Municipality (21230711000) and the General
Fund of Natural Science Foundation of China (42275124). Yong Jie Li acknowledges the financial support from Science and
Technology Development Fund, Macau SAR (File No. 0031/2023/AFJ and 0107/2023/RIA2) and multiyear research grants
(No. MYRG-GRG2023-00008-FST-UMDF and MYRG-GRG2024-00032-FST-UMDF) from the University of Macau.

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
