# Peer review of "A Comprehensive Characterization of Empirical Parameterizations"

_EGUsphere, 2024_

## Author Comment (AC1)

We sincerely thank the editor and reviewers for the valuable comments and suggestions, which help improve the manuscript. We herein provide the point-by-point response and the changes made to the manuscript. The response is in the indent and blue, and the revised text is in the indent and green. The line numbers mentioned in the response correspond to the revised manuscript unless otherwise specified.

*Reviewer #1:*

*General comment:*

*Liu et al. explicitly examined the OH exposure quantification in PAM-OFR by comparing the results of calibration experiments with a limited set of calibration conditions and a wide range of calibration conditions. Recommendations and cautions have been given for both modes of PAM-OFR when calibrating OH exposures in the laboratory. This paper has important implications for air quality and atmospheric chemistry studies considering the increasing number of PAM-OFR users in the community. I recommend its publication after the following minor comments are addressed.*

Response: We thank the reviewer for the positive comments. We address the questions and comments as follows.

*Specific comments:*

*1. Lines 121-122: Is this statement also true for other OFRs with different UV lamps and different designs of reactors (e.g. wall material, shape, or volume)?*

Response: We appreciate the reviewer's comment. As stated in lines 197-199 and lines 410-413 of the manuscript, the generation of OH radicals in PAM-OFR is related to the photon fluxes at $\lambda = 185$ nm ($I_{185}$) and $\lambda = 254$ nm ($I_{254}$). According to Rowe et al. (2020), $I_{185}:I_{254}$ is specific to the Hg lamp utilized. The conclusions drawn about the OFR185 mode are valid only if one already has a set of $a$–$f$ values appropriate for the specific UV lamps being used, as the $I_{185}:I_{254}$ ratio affecting the $OH_{exp}$ is lamp-specific. For a PAM-OFR employing a different Hg lamp, a series of calibration experiments should be conducted in any case.

Different designs of reactors (e.g. wall material, shape, or volume, etc.) can lead to differences in the actual distribution of reaction times, heterogeneity in the UV light intensity and the concentration of radicals/oxidants at different points within the reactor, and wall reactions. When Li et al. (2015) and Peng et al. (2015) modeled the radical chemistry in the PAM-OFR and developed the $OH_{exp}$ estimation equation, they assumed the use of plug flow conditions, which neglected the differences mentioned above.

Moreover, the OH exposure values from the plug flow model results have been shown by them to have no significant difference from the results calculated using the residence time distribution (RTD) in the model. Therefore, for oxidation flow reactors other than the PAM-OFR, if the assumption of plug flow conditions is valid, then the conclusions stated in lines 121-122 should hold.

To make it clearer, we modified this part in the revised manuscript as follows:

Line 121-122: "The methodology of this study can be applied to laboratory and field experiments for $OH_{exp}$ estimation using PAM-OFR or other OFRs that follow a plug-flow assumption."

2. *Line 145: Please check if 0.1 ppm is a typo because 0.1 ppm of hydrocarbon seems*

*high.*

Response: Thanks for pointing this out. We have corrected it accordingly.

Lines 144-145: "(2) dry clean air from a zero-air generator (ZAS-100/150,

Convenient) with a non-methane hydrocarbon content of less than 1 ppb;"

3. *Line 231: What does a slight change of residence time mean here? 5% variation?*

Response: Thanks for the valuable comment. Rowe et al. (2020) fit a set of parameters *a–f* for the $OH_{exp}$ estimation equation in the OFR185 mode as follows:

$a = 10.098$, $b = 0.15062$, $c = -0.44244$, $d = 0.18041$, $e = 0.031146$, and $f = 0.1672$.

These parameters were derived under the following experimental conditions: (1)

residence time: 124 s; (2) water vapor mixing ratio: 0.1–3 %; (3) external OH

reactivity: 0.77–232 s$^{-1}$; (4) photon flux at $\lambda$ = 185 nm ($I_{185}$): $10^{13}$–$10^{16}$ photons cm$^{-2}$ s$^{-1}$; (5) $I_{185}$:$I_{254}$ = 0.066.

Avery et al. (2023) used the same set of parameters to estimate OH$_{exp}$ under the following experimental conditions: (1) residence time: 130 s; (2) water vapor mixing ratio: 1.5 % (calculated from the given average relative humidity of 40 % and average temperature of 28°C); (3) external OH reactivity: 0.6 s$^{-1}$; (4) photon flux: $1 \times 10^{14}$–$3 \times 10^{15}$ photons cm$^{-2}$ s$^{-1}$ (it was not specified whether this photon flux was for the 185 nm or 254 nm UV lamp); (5) $I_{185}$:$I_{254}$ = 0.066.

The residence time differed by 4.8 %, but the OHR$_{ext}$ varied a lot in Rowe et al. (2020). Therefore, the conclusion in Avery et al. (2023) that "The estimated uncertainty in calculated OH$_{exp}$ values was ± 50 %." might originate primarily from the difference in OHR$_{ext}$, instead of residence time. To avoid misunderstanding, this sentence in line 231 is deleted in the revised manuscript.

*4. Line 281: Please define FP$_{eOHR, 185}$.*

Response: Thanks for pointing this out. This has been revised.

Line 281: "We then investigated the possible causes of the discrepancy for OH$_{ext}$ estimation between FP$_{lOHR, 185}$ and FP$_{eOHR, 185}$ (eOHR: extended OHR$_{ext}$)."

*5. Also, regarding the discrepancy for OH estimation between low OHR and extended high OHR, would the oxidation of SO$_2$ by H$_2$O$_2$ in nucleated sulfuric acid aerosols contribute to such discrepancy as H$_2$O$_2$ would be formed in the OFR and could further oxidize SO$_2$ in the aqueous sulfuric acid aerosols (Liu et al., 2020)?*

Response: We thank the reviewer for the suggestion and agree that this could be one of the reasons that cause the discrepancy in OH estimation between low OHR and extended high OHR. We revised and supplemented the manuscript as suggested:

Line 282: "From a mathematical perspective, according to Eq. 2, the third term $c \times OHR_{ext}^d \times \log(O_{3, \, out} \times 180/t)$ and the fourth term $e \times OHR_{ext}^f \times [\log(O_{3, \, out} \times 180/t)]^2$ are associated with OHR$_{ext}$, which involve fitted parameters of $c$–$f$."

Lines 300-302: "From the perspective of oxidation chemistry, high concentrations of gas phase $SO_2$ could lead to more $SO_2$ entering the particle phase. The $H_2O_2$ in the liquid water of nucleated sulfuric acid aerosols would further oxidize $SO_2$ (Liu et al., 2020) which could lead to the discrepancy for OH estimation between low OHR and extended high OHR."

6. *The data points for OFR254 mode are more scattering. Are there any recommendations to improve the OH exposure estimation for the OFR 254 mode?*

Response: Thanks for the comment. As described in lines 439-441 of the manuscript, there are two pathways for the generation of OH radicals in the OFR185 mode, while the OFR254 mode has a lower capacity for generating OH radicals. This may be the reason why it is less accurate in obtaining OH exposure through estimation equations that require [OH] to be in a steady state. At present, we recommend avoiding the use of too-low lamp power settings when using the OFR254 mode (as mentioned in lines 395-396), to minimize the occurrence of significantly deviated estimation results. However, there is currently no good suggestion for improving the estimation level of OFR254 mode to be similar to that of OFR185 mode.

*Reviewer #2:*

*General comment:*

*This study assesses the accuracy of empirical equations for estimating hydroxyl*
*radical exposure ($OH_{exp}$) in an oxidation flow reactor (OFR) under varied experimental*
*conditions. Through 62 experiments, it was found that parameters derived from narrow*
*calibration ranges, such as water vapor mixing ratios or low external OH reactivity,*
*can be applied in broader scenarios for OFR254 mode but show limitations in OFR185*
*mode at high external OH reactivity. The authors found that at least 20–30 data points*
*are necessary to derive reliable parameters. The findings highlight the need for caution*
*when extrapolating parameters to conditions beyond their calibration range,*
*particularly for scenarios involving high OH reactivity.*

*This study is planned well and the scope of having better characterized*
*experiments when using OFRs is important. Furthermore, the manuscript is, for most*
*parts, written well and clearly.*

*Although I have some comments that should be addressed before this study can be*
*published.*

Response: We thank the reviewer for the positive comments. We address the
questions and comments as follows.

*Specific comments:*

1. *Lines 20: I would not suggest to introduce the $OH_{exp}$, which is a main concept of*
*that paper, in brackets. Change sentence to clarify.*

Response: We thank the reviewer for the suggestion. We have modified this
sentence as suggested in the revised manuscript as follows:

Line 20: "OH exposure ($OH_{exp}$), representing the extent of hydroxyl radical (OH)
oxidation and normally expressed as the product of OH concentration and residence
time in the OFR, is important in assessing the oxidation chemistry in SOA formation."

2. *Line 33: Give also RH values (at standard conditions), because a lot other studies*

*use that, just for comparisons..*

Response: Thanks for the valuable suggestion. We have revised as suggested:

Line 32-35: "For example, parameters derived within a narrow water vapor mixing ratio range (0.49–0.99 %, corresponding to 15.1–30.8 % of relative humidity at 101.325 kPa and 298 K) can be extended to estimate the $OH_{exp}$ under the entire range of water vapor mixing ratios (0.49–2.76 %, equivalent to 15.1–85.7 % of relative humidity under identical conditions)."

3. *Line 39: "but too-low lamp voltages should be avoided." For clarification add why.*

Response: Thank you for the suggestion. We have modified this part as suggested:

Lines 37-40: "For the OFR254 mode (254-nm lamps with external $O_3$ generation), all parameters obtained within a narrow range of conditions can be used to estimate $OH_{exp}$ accurately when experimental conditions are extended. Additionally, when using the OFR254 mode, too-low lamp voltages should be avoided, as they will generally result in large deviations in the estimations of $OH_{exp}$ from empirical equations."

4. *Sentence starting at line 77 and general comment for introduction: It is definitely an argument that the PAM-OFR is widely used, but I find you should mentioned some other OFRs and their advantages and disadvantages and why you chose this one (you have some citations at line 54, but I think it is not enough to just cite them there). Almost all other OFRs have also published method paper where at least some of the aspects you discuss in this paper have also already been measured and you should cite them and describe the existing literature better. In line 122 you say other OFRs but without having a paragraph introducing them, it is unclear for the reader why this should be the case.*

Response: Thank you for the suggestion. We have modified the introduction to more clearly introduce other OFRs along with their advantages and disadvantages, while also explaining why we chose to study the PAM-OFR. The revised manuscript is shown below:

Lines 54-66: "Smog chambers (Cocker et al., 2001; Hildebrandt et al., 2009; Wang et al., 2014) and oxidation flow reactors (OFRs) (George et al., 2007; Kang et al., 2007; Lambe et al., 2011) have been widely employed to simulate oxidation of VOCs and subsequent SOA formation. For example, the Caltech Chamber provides oxidation conditions close to the real atmosphere, making it suitable for the study of complex multi-step reactions and low-volatility products. However, each experiment takes several hours to days and long-duration experiments are prone to background interference. The Toronto Photo-Oxidation Tube (TPOT) focuses on the study of heterogeneous oxidation reactions of aerosols. Its 0.8 L volume makes it portable, but it is prone to uneven residence time distribution (RTD) and significant wall effects. The Potential Aerosol Mass Oxidation Flow Reactor (PAM-OFR) and the Gothenburg Potential Aerosol Mass Oxidation Flow Reactor (Go: PAM-OFR) are often used to study the transformation of gaseous precursors into particles (such as the formation of SOA). The Go: PAM-OFR has a volume of 7.2 L, which is only half that of the PAM-OFR, making it suitable for experiments on mobile platforms. However, its small volume gives it the same disadvantages as the TPOT, and it is equipped with only a single UV lamp, which does not allow for as wide a range of controllable oxidation levels as the PAM-OFR. The PAM's moderate volume and central flow sampling can reduce wall effects.

These OFRs normally operate with high concentrations of oxidants (e.g., OH radicals), which lead to a significant acceleration of oxidation reactions, often by orders of magnitude. … Accurate measurement or estimation of the $OH_{exp}$ during laboratory experiments, therefore, is the key to understanding the oxidation chemistry that can represent the ambient conditions. In this study, we chose to further investigate the PAM-OFR to explore its $OH_{exp}$, as it offers moderate conditions in terms of experiment time, deployment complexity, range of oxidation levels, and wall effects."

Response: Since the definition of RTD has been given in the revised text above, it will be written directly as RTD in line 163.

Line 163: "…which has been shown to agree with the RTD approach for $OH_{exp}$…"

5.  *Line 100: "but this requires specific instruments» specify which instruments.*

    Response: Thanks for pointing this out. In the revised manuscript, we clarified that these organic compounds need to be measured by PTR-TOF-MS or PTR-MS. Additionally, based on Comment #7, we supplemented information on other methods currently being used to indirectly quantify OH radical concentrations by measuring the decay of tracers such as 3-pentanol, 3-pentanone, pinonaldehyde, and butanol-d9.

    Line 99-104: "In some field studies using PAM-OFR, concurrent $OH_{exp}$ was estimated by measuring the relative decay of benzene and toluene (Liao et al., 2021; Liu et al., 2018). Additionally, some studies have mentioned that OH concentrations can be indirectly measured by detecting the decay of tracers such as 3-pentanol, 3-pentanone, pinonaldehyde, or butanol-d9 (Barmet et al., 2012). However, the measurement of all these organic tracers requires specific, sophisticated instruments such as proton-transfer-reaction time-of-flight mass spectrometers (PTR-MS)."

6.  *Paragraph 2.1 (and2.2) A schematic would help to better understand the setup and which instruments were used.*

    Response: Thank you for the suggestion. In order not to occupy space in the main text, the schematics of the PAM-OFR experimental setup and the examples of a set of experiments conducted in OFR185 mode and OFR254 mode are shown in Figures S1 and S2 in the Supplementary Information, respectively.

7.  *Line 114: Again maybe also mentioned other method (like deuterated butanol etc.) that other people are using.*

    Response: Thank you for the valuable suggestion. The modifications and additions to this part have been responded to in Comment #5.

8.  *Line 123 Methods: Have you considered temperature in the OFR? If so please state in the manuscript and if not I would highly suggest to do so. We saw that the*

*temperature inside the OFR (which is not constant especially for 185 and 254 mode due to different power consumption) makes a massive difference in the OH concentration and that this is an important parameter that needs to be taken into account.*

Response: Thank you for the valuable comment. We would like to stress that we introduced a nitrogen purge gas at a flow rate of 0.2 to 0.3 L min$^{-1}$ between the lamp and the sleeve to reduce the heat generated by the lamps. This reduced the heating inside the PAM-OFR but still resulted in a maximum deviation of ~13 °C (with $SO_2$ as the OHR source). With this maximum temperature deviation included, ~10 % higher than that calculated using the rate constant at 25 °C was estimated for oxidation of $SO_2$ by OH radicals as estimated by the Arrhenius equation. Pan et al. (2024) noted that temperature increases caused by lamp heating exerted minimal influence on gas-phase reaction rates, with $SO_2$ decay and OH exposure showing negligible variations. Consequently, we did not consider the impact of temperature on reaction kinetics in this study.

9. *Methods: Add a small paragraph with chemicals (purity) and gases that you used.*

Response: Thank you for the suggestion. The purity of $SO_2$ and CO standard gases has been added in the revised manuscript as shown below:

Lines 142-144: "The trace-gas flow, i.e. $SO_2$ of 0.2–8.7 ppm or CO of 10.2–207.5 ppm supplied from gas cylinders (Purity: 99.9 % of $SO_2$, 99.95 % of CO; Shanghai Shenkai Gases Technology CO., LTD.);"

10. *Line 154 Clarify why "allowed to stabilize".*

Response: Thank you for the suggestion. We have modified this part as suggested:

Lines 153-154: "In the OFR185 mode, without radical generation to oxidize the tracer species, their concentration was allowed to stabilize under dark conditions.

11. *Line 176 and onward: It is confusing that a-f and a-c have same characters for different equations. I would suggest to use different letters for clarification.*

Response: Thank you for the suggestion. To better distinguish the fitting parameters in Eq. 2 and Eq. 3, we have changed the parameters $a$–$c$ in Eq. 3 to $x$–$z$ in the revised manuscript and the Supplementary Information.

Line 95: "…($a$–$f$ and $x$–$z$, respectively)…"

Line 182: "$OH_{exp,\ est} = 10^{\left[x+\log(-\log r_{O_3})+y\times\left(\frac{OHR_{ext}}{O_{3,\ in}}\right)^z\right]}$,"

Line 193: "we used Eqs. 2 and 3 to fit the parameters $a$–$f$ and $x$–$z$ for OFR185 and OFR254 modes, respectively…"

Line 202: "Similarly, fitting parameters $x$–$z$ for OFR254 mode…"

Line 364: "The parameters $x$–$z$ ($FP_{lOHR,\ 254}$; lOHR: low external OHR) (Table S4) were obtained by fitting Eq. 3 to $OH_{exp,\ dec}$."

Line 381: "…using the fitting parameters ($x$–$z$) obtained from a narrower range of data points."

Line 387: "…were used to re-fit the parameters $x$–$z$,…"

Lines 408-409: "…can be used to estimate $OH_{exp}$ reasonably well using the parameters $x$–$z$ obtained from a narrower range of data points."

The revisions in parameters $a$–$c$ to $x$–$z$ in the Supplementary Information are not listed here but have been updated in the revised Supplementary Information.

12. *Line 209: SO₂ not defined, explain how SO₂ used as OHR$_{exp}$. This also needs to be done in the method part.*

Response: Thanks for pointing this out. We have added the definitions of $SO_2$ and CO to their first occurrence in the revised manuscript, i.e. line 38.

Lines 40-41: "…, at least 20–30 data points from sulfur dioxide ($SO_2$) or (carbon monoxide) CO decay with varying conditions…"

To explain more clearly how $SO_2$ was used as $OHR_{ext}$ and used to measure $OH_{exp}$, we modified and supplemented the "2 Method" section as follows:

Line 137: "$OH_{exp}$ can be indirectly measured by detecting the decay of the tracers with known reaction rates."

*13. Figure 1:(and other figures) add error bars or at least explain why the data points*

*don't have error bars.*

Response: Thanks for the comment. In our study, the error values for all $OH_{exp, dec}$ values are one-half or even two orders of magnitude smaller than the respective $OH_{exp, dec}$ values. When a logarithmic scale is applied to the X-axis, the error bars become difficult to represent. To enhance the readability of the figure, we have not included error bars in the graph. For the same reason, we have also chosen not to display the error bars of $OH_{exp, est}$ values on the Y-axis.

*Explain 1:2 and 2:1 line in the caption and in the main text why you used this.*

Response: We chose the 1:2 and 2:1 lines to indicate approximately half an order of magnitude difference between $OH_{exp, dec}$ and $OH_{exp, est}$, which is considered to be acceptable as an uncertainty in $OH_{exp}$ estimation.

Line 212-213: "The 1:2 and 2:1 lines indicate approximately half an order of magnitude difference between $OH_{exp, dec}$ and $OH_{exp, est}$, which is considered to be acceptable as an uncertainty in $OH_{exp}$ estimation."

*14. Figure 1: Figure 1 is very packed. It is hard to read the photochemical age in the*

*last row and vice versa for $OH_{exp, dec}$. Maybe you can think of making this figure a*

*little bit less crowded… Three single figures? Also it would help if you remove the*

*black lines around the data point and make them a little transparent.*

Response: Thank you for the suggestion. We have modified the display of the photochemical age in Fig. 1 to enhance readability. Figs. 2-5 and Fig. S3 have also undergone similar modifications in the revised manuscript, which are not displayed here. For the X-axis of $OH_{exp, dec}$, we aimed to use the same range for all figures utilizing $OH_{exp, dec}$ as the X-axis. To ensure that all data points in Figure 3c are visible, we also selected this range for Figure 1, without considering a zoom-in approach to make the data points look more scattered. Removing the black lines surrounding the data points would make the points color-coded with the same or similar colors indistinguishable and overlap into one piece. Therefore, we chose to retain the black outlines.

[Figure]

Figure 1: The regression results of $OH_{exp, est}$ and $OH_{exp, dec}$ when variations occurred in (a1–a3) residence time, (b1–b3) water vapor mixing ratio, and (c1–c3) output $O_3$ concentration under atmospheric relevant $OHR_{ext}$ level (4–23 s$^{-1}$). Compared to panels a1, b1, and c1, panels a2, b2, and c2 respectively incorporated additional data points with higher t, $H_2O$, and $O_{3, out}$ values, but still utilized the fitting parameters $FP_{st, 185}$, $FP_{IH2O, 185}$, and $FP_{IO3, 185}$ obtained from the lower condition range to estimate $OH_{exp, est}$. In panels a3, b3, and c3, all data points within the extended condition range were used to re-fit the parameters $a$–$f$, and the resulting $FP_{et, 185}$, $FP_{eH2O, 185}$, and $FP_{eO3, 185}$ were employed to estimate $OH_{exp, est}$ (s: short, l: low, e: extended).

*15. Line 275: $FP_{IOHR}$ not defined. In general, I would recommend to overthink your abbreviations. I guess, for you they all make completely sense, but for an outside reader it is very confusing, also because they are very similar. You could make the paper more "reader-friendly" if you improve that. Having more than one information in the subscript is also not recommended. As example write instead of $OH_{exp, est}$ the estimated $OH_{exp}$. It is a technical journal not one where you have a*

*strict word limit, so you should invested a little bit more in clarity of the technical*

*details.*

Response: Thanks for the comment. $FP_{lOHR}$ has been defined in line 275 in the manuscript as "the parameters $a$–$f$ ($FP_{lOHR, 185}$; lOHR: low $OHR_{ext}$)". We have defined all abbreviations and subscripts in the article in our manuscript. To keep the main text from becoming overly lengthy, we use the current forms of abbreviations and subscripts.

*16. Figure 2: Explain red "outliers" in (b).*

Response: Thanks for the comment. The red "outliers" in Figure 2b were explained in lines 276-280 in the manuscript: "It could be observed from Figure 2b that when estimating $OH_{exp}$ using $FP_{lOHR, 185}$, $OH_{exp, est}$ of the high-$OHR_{ext}$ data points were significantly overestimated, with a difference of more than two orders of magnitudes compared to $OH_{exp, dec}$. This observation suggests that different from cases for residence time, water vapor mixing ratio, and ozone concentration shown in the section above,

$FP_{lOHR, 185}$ were not applicable to high-$OHR_{ext}$ conditions."

To help readers better understand the information, we have supplemented the figure caption for Figure 2 as follows:

Lines 313-315: "…but $FP_{lOHR, 185}$ were still used to estimate $OH_{exp, est}$. In panel b, data points in red showed that the $OH_{exp, est}$ of these high-$OHR_{ext}$ data points were significantly overestimated. $FP_{lOHR, 185}$ were not applicable to high-$OHR_{ext}$ conditions.

The data points in panel c…"

*17. Line 372: Clarify why not suitable.*

Response: Thanks for the comment. First, in terms of light intensity, under the same lamp power, the ozone-producing Hg lamps (GPH436T5VH/4P) used in the

OFR185 mode and the ozone-free mercury lamps (GPH436T5L/4P) used in the

OFR254 mode output the same light intensity at 254 nm (https://www.light- sources.com/solutions/germicidal-uvc-lamps/uv-germicidal-lamps/low-pressure- mercury-lamps/254-nm-uv-lamps/). However, the VH (Very High ozone-producing)

lamps can produce energy at both the wavelength of 254 nm and 185 nm. Secondly, regarding the energy of individual photons, shorter wavelength light possesses higher photon energy. The PAM-OFR system generates OH radicals through the photolysis of water and ozone (the OFR25 mode only photolyzes ozone). Therefore, overall, the ability of the OFR185 mode to produce OH radicals is greater than that of the OFR254 mode, leading to a higher achievable $OH_{exp}$. Consequently, the OFR254 mode is less suitable than the OFR185 mode for studying oxidative chemistry under high $OHR_{ext}$ conditions.

We also want to modify a typo as follows:

Lines 129: "…OFR254 using two ozone-free Hg lamps (GPH436T5L/4P, Light Sources, Inc.) …"

*18. Line 395: Specify a threshold, "too low" is not very scientific.*

Response: Thank you for the suggestion. We have modified this part as follows:

Lines 395-396: "…too low lamp power settings, for example, UV lamp voltage below 1.5V should be avoided in the case of our study."

*19. Line 420: This paragraph contains methods and new results, so please move it up. Also figure 6 should be results. A conclusion should conclude and not show new results if possible.*

Response: Thanks for the comment. The previous results primarily focus on investigating the impact of varying conditions on the applicability of parameters under two different modes. In the paragraph starting from line 420, we do not introduce new data results; rather, we employ a random sampling method to estimate how many experimental data points we need to obtain the reliable $OH_{exp}$ estimation. This serves as further exploration based on the previous results, making it an implication and suggestion. Therefore, we prefer to put it to the conclusion.

*20. Figure 6: use scatter plot with no lines, maybe then it is possible to see the data better. Also second Y-axis is different, use the same numbers for better comparison. (same for third Y-axis). Also some of the error bars are outside of the graph. Show all the data or at least justify why you don't show all the error bars. Explain why some of them are very big, especially in the beginning.*

Response: Thanks for the valuable suggestion. We have modified Figure 6 to be a scatter plot without a line. The second and third Y-axes of Figure 6 have also been modified as suggested for better comparison. To ensure that readers with color vision deficiencies can correctly obtain the information, we also changed the color scheme used in our figure, which is shown below:

[Figure]

Initially, when the number of selected data points is insufficient, the fitted parameters $a$–$f$ or $x$–$z$ exhibit high inaccuracy in estimating $OH_{exp, est}$, resulting in a significant deviation from $OH_{exp, dec}$. Therefore, in the beginning, the error bars for R-squared values, slopes, and intercepts were quite large. By observing the stability of the R-squared values, slopes, and intercepts, as well as the reduction in their error bars, we can infer the minimum number of decay experiments required to obtain reliable $OH_{exp, est}$. The reason why not all error bars were shown is that in some cases the results from ten attempts were highly inconsistent, leading to large error bars. If they were fully shown, it would be difficult to see other data points and information in Figure 6. Our primary interest is in determining the minimum number of decay experiments needed for reliable $OH_{exp, est}$, rather than focusing on the extent of deviation resulting from insufficient experimental quantities. Therefore, this information is not fully presented in Figure 6.

*21. Conclusion Try to give an outlook on how your findings can be used with other systems. This would improve the quality of the manuscript so that also "non PAM-OFR" users could profit from you work..*

Response: Thanks for the valuable suggestion. We have made the following additions in the conclusion to provide some suggestions for other OFR users:

Line 447-452: "…using the OFR254 mode is a better choice. For users of other OFRs (non-PAM-OFR) who would like to apply the conclusions above, at least two conditions must be met: (1) the concentration of [OH] within the OFR should remain stable, and (2) the assumption of plug flow conditions is acceptable, allowing for the neglect of differences in the actual RTD, heterogeneity in the UV light intensity and the concentration of radicals/oxidants at different points within the reactor, which are caused by different designs of reactors (such as wall materials, shapes, or volumes)."

*Technical comments:*

*1. Sentence starting at line 30 need clarification.*

Response: Thanks for the comment. The detailed explanation of "We showed that, ... in wider ranges of conditions." was given in the next sentence, which is "For example, parameters derived within a narrow water vapor mixing ratio range (0.49–0.99 %, corresponding to 15.1–30.8 % of relative humidity at 101.325 kPa and 298 K) can be extended to estimate the $OH_{exp}$ under the entire range of water vapor mixing ratios (0.49–2.76 %, equivalent to 15.1–85.7 % of relative humidity under identical conditions)." in lines 32-35.

*2. Lines 84-90 cite equations.*

Response: Thanks for the suggestion. We have added the citation of Equations R1–R6 in line 84 as follows:

Lines 83-84: "…or photolysis of externally added $O_3$ at $\lambda$ = 254 nm (OFR254; R5–R6) (Rowe et al., 2020):"

3.  *Line 258: OFR254 is not defined. It is clear from the context, but I would suggest*
    *to define it the first time you use it.*

    Response: Thanks for the comment. The definition of OFR254 has been given in
    line 37 of the manuscript: "…the OFR254 mode (254-nm lamps with external $O_3$
    generation),…"

4.  *Line 281: OHext is not defined.*

    Response: Thank you for pointing out this mistake. We have corrected the typo.

    Line 281: "We then investigated the possible causes of the discrepancy for $OH_{exp}$
    estimation between $FP_{lOHR, 185}$ and $FP_{eOHR, 185}$."

*Reviewer #3:*

*General comment:*

*This article evaluates the impact of various factors on estimated OH exposure*

*($OH_{exp}$) under the OFR185 and OFR254 operating modes of the PAM-OFR through a*

*series of experiments. It provides valuable guidance for the broader application of OFR*

*and the comparability of their results. However, several critical issues need to be*

*addressed before publication.*

*$OH_{exp}$ can be divided into offline calibration and online calibration. The method*

*introduced in this paper, which derives estimation equations through a series of*

*univariate controlled experiments, belongs to offline calibration. On the other hand,*

*the estimation of $OH_{exp}$ during field experiments, which involves observing the decay*

*of highly reactive precursors inside the OFR, constitutes online calibration. The*

*starting point of this study is to determine the minimum number of experiments required*

*to obtain accurate offline calibration equations for $OH_{exp}$. However, several issues need*

*to be addressed:*

Response: We thank the reviewer for the positive comments. We address the questions and comments as follows.

1. *Priority of Online Calibration: If VOC concentrations before and after the OFR*

*can be measured online during field observations, this should be the preferred*

*approach. Offline calibration cannot simulate all real-world conditions, so the*

*applicability of the offline method should be clarified.*

Response: Thanks for the valuable suggestion. As the reviewer mentioned, offline calibration cannot simulate all real-world conditions. Measuring VOC

concentrations online before and after the OFR during field observations to calculate

$OH_{exp}$ would be a better choice. However, this also presents some inconveniences, such as the need to switch the instrument used for measuring VOC concentrations (e.g., PTR-TOF-MS) back and forth between the front and end of the OFR, which can result in some loss of real-time data for the VOCs before entering the OFR during the field study. Therefore, we hope to estimate $OH_{exp}$ using an empirical equation composed of parameters that are easy to measure in real-time (Eqs. 2 or 3), allowing us to capture real-time OH exposure. To clarify our purpose in exploring the applicable conditions for the estimation equation, we have made the following modifications in the revised manuscript.

Line 99-105: "In some field studies using PAM-OFR, concurrent $OH_{exp}$ was estimated by measuring the relative decay of benzene and toluene (Liao et al., 2021; Liu et al., 2018). Additionally, some studies have mentioned that OH concentrations can be indirectly measured by detecting the decay of tracers such as 3-pentanol, 3-pentanone, pinonaldehyde, or butanol-d9 (Barmet et al., 2012). However, the measurement of all these organic tracers requires specific, sophisticated instruments such as proton-transfer-reaction time-of-flight mass spectrometers (PTR-MS). Additionally, switching the instrument back and forth between the front and end of the OFR during field measurements can result in some loss of real-time VOCs data before entering the OFR."

2. *$OH_{exp}$ Range: Due to its portability, the OFR has significant advantages in field observations and is often used to simulate VOC oxidation and subsequent SOA formation under high OH exposure conditions. Therefore, the calibration experiments should cover the typical $OH_{exp}$ range used in field studies, potentially extending to several days. However, some single-variable experiments in the paper only achieve a maximum $OH_{exp}$ of 1 day. Additionally, readers are likely to be interested in the specific error values under different $OH_{exp}$ conditions, so quantitative results should be provided.*

Response: Thanks for the valuable suggestion. In the revised manuscript, we provide the range of $OH_{exp}$ covered in the experiments conducted under OFR185 mode and OFR254 mode in our study as follows:

Line 186-190: "We have performed in total of 62 sets of trace-gas decay experiments with 416 data points for the $OH_{exp, dec}$, with 25 sets and 175 data points in the OFR185 mode and 37 sets and 241 data points in the OFR254 mode. In OFR185 mode, the 175 experiments cover an $OH_{exp, dec}$ range of $3.6 \times 10^8$–$5.5 \times 10^{12}$ molecules $cm^{-3}$ s, with an equivalent photochemical age ranging from 4 minutes to 43 days. In OFR254 mode, the 241 experiments cover an $OH_{exp, dec}$ range of $1.01 \times 10^9$–$2.18 \times 10^{12}$ molecules $cm^{-3}$ s, with an equivalent photochemical age ranging from 11 minutes to 17 days."

For specific error values under different $OH_{exp}$ conditions,

Lines 190-192: "The error in $OH_{exp, dec}$ is derived from the measurement error of the tracer gas, propagated through Eq. 1. When $OH_{exp, dec}$ ranged from $3.6 \times 10^8$–$5.5 \times 10^{12}$ molecules $cm^{-3}$ s, the resulting error values were $1.9 \times 10^8$–$2.4 \times 10^{10}$ molecules $cm^{-3}$ s."

Lines 196-197: "…via linear regression analysis. Similarly, the error values for all $OH_{exp, est}$ values are at least one order of magnitude smaller than the respective $OH_{exp, est}$ values. The generation of OH radicals…"

3. *Exclusion of Irrelevant Experiments: In field observations, the OFR residence time is generally fixed, while other conditions, such as RH, may vary with the environment. When calculating the minimum number of experiments, were experiments that altered residence time excluded? Under OFR254 conditions, if UV lamp pressure and RH are kept constant while only the $O_3$ concentration entering the reactor is varied, would it still be possible to achieve different $OH_{exp}$? If so, can the number of experiments be further minimized? This paper should provide guidance on which variables should be set within a wider range and which variables only need to be adjusted within a narrower range, rather than prescribing a specific number of experiments.*

Response: Thanks for the valuable suggestion. The estimation of the minimum number of experiments was done by merely random sampling of the available data in our data sets. It gives a rough estimation of reasonable time/data points to obtain reliable $OH_{exp}$ through fitting the equations, instead of covering all experimental conditions. We therefore do not intend to remark on too generalized guidance on this.

*4. Variation Ranges of Other Factors: The description of the variation ranges of other*

*factors in the experiments investigating their effects on $OH_{exp}$ estimation is unclear.*

*This information needs to be added.*

Response: Thanks for the valuable suggestion. We have added Tables S3 and S4

in the revised manuscript of the main text and Supplementary Information to show the range of all experimental conditions for the different datasets used to fit the parameters *a–f* and *x–z* and evaluate their applicability, respectively.

Lines 215-216: "…and also a range of higher values (61–200 s). The detailed ranges of each experimental condition for different datasets are listed in Table S3. With the residence time of 33 s,…"

Lines 249-250: "…where the narrow range was situated within the higher interval.

The detailed ranges of each experimental condition for different datasets are listed in

Table S3. As shown in Figure S3,…"

Lines 359-360: "…the three parameters potentially affecting the $OH_{exp}$ are $OHR_{ext}$, input $O_3$ concentration, and $r_{O3}$. The detailed ranges of each experimental condition for different datasets are listed in Table S4. We found that compared to Figure 1,…"

Supplementary Information:

**Table S3:** In OFR185 mode, the range of various experimental conditions involved in
the different datasets when fitting parameters *a–f* and evaluating their applicability.

| Figure # | Residence time (s) | Water vapor mixing ratio (%) | Output $O_3$ concentration (molecules cm$^{-3}$) | External OHR (s$^{-1}$) | OHR source |
|---|---|---|---|---|---|
| Fig. 1a1 | 33 | 0.54–1.48 | $1.44 \times 10^{12}$–$1.89 \times 10^{14}$ | 4–23 | $SO_2$ |
| Fig. 1a2 | 33–200 | | | | |
| Fig. 1a3 | | | | | |
| Fig. 1b1 | 33–200 | 0.49–0.99 | $1.44 \times 10^{12}$–$2.03 \times 10^{15}$ | 4–18 | $SO_2$ |
| Fig. 1b2 | | 0.49–2.76 | | | |
| Fig. 1b3 | | | | | |

| Figure # | Residence time (s) | Water vapor mixing ratio (%) | Output O$_3$ concentration (molecules cm$^{-3}$) | External OHR (s$^{-1}$) | OHR source |
|---|---|---|---|---|---|
| Fig. 1c1 | | | $1.44 \times 10^{12}$–$6.79 \times 10^{13}$ | | |
| Fig. 1c2 | 33–296 | 0.49–1.62 | $1.44 \times 10^{12}$–$2.03 \times 10^{15}$ | 4–23 | SO$_2$ |
| Fig. 1c3 | | | | | |
| Fig. 2a | | | | 4–23 | |
| Fig. 2b | 33–296 | 0.49–2.76 | $1.44 \times 10^{12}$–$2.03 \times 10^{15}$ | 4–204 | SO$_2$ |
| Fig. 2c | | | | | |
| Fig. 3a | 33–296 | 0.38–2.76 | $1.44 \times 10^{12}$–$2.03 \times 10^{15}$ | 4–204 | SO$_2$ |
| Fig. 3b | 33 | 0.85–1.17 | $1.95 \times 10^{12}$–$1.88 \times 10^{14}$ | 61–1227 | CO |
| Fig. 3c | 33–296 | 0.38–2.76 | $1.44 \times 10^{12}$–$2.03 \times 10^{15}$ | 4–1227 | SO$_2$ + CO |
| Fig. S3a1 | 100–296 | | | | |
| Fig. S3a2 | 33–296 | 0.63–2.76 | $8.16 \times 10^{12}$–$2.03 \times 10^{15}$ | 6–18 | SO$_2$ |
| Fig. S3a3 | | | | | |
| Fig. S3b1 | | 1.04–2.76 | | | |
| Fig. S3b2 | 33–296 | 0.49–2.76 | $3.31 \times 10^{12}$–$1.16 \times 10^{15}$ | 8–23 | SO$_2$ |
| Fig. S3b3 | | | | | |
| Fig. S3c1 | | | $8.45 \times 10^{13}$–$2.03 \times 10^{15}$ | | |
| Fig. S3c2 | 33–296 | 0.50–2.76 | $1.44 \times 10^{12}$–$2.03 \times 10^{15}$ | 4–23 | SO$_2$ |
| Fig. S3c3 | | | | | |

**Table S4:** In OFR254 mode, the range of various experimental conditions involved in
the different datasets when fitting parameters $x$–$z$ and evaluating their applicability.

| Figure # | External OHR (s$^{-1}$) | Output O$_3$ concentration (molecules cm$^{-3}$) | $r_{O3}$ | OHR source |
|---|---|---|---|---|
| Fig. 4a1 | 5–14 | | | |
| Fig. 4a2 | 5–21 | $6.46 \times 10^{13}$–$4.77 \times 10^{14}$ | 0.42–1.00 | SO$_2$ |
| Fig. 4a3 | | | | |
| Fig. 4b1 | | $6.46 \times 10^{13}$–$1.62 \times 10^{14}$ | | |
| Fig. 4b2 | 5–21 | $6.46 \times 10^{13}$–$4.77 \times 10^{14}$ | 0.66–1.00 | SO$_2$ |
| Fig. 4b3 | | | | |
| Fig. 4c1 | | | 0.69–0.90 | |
| Fig. 4c2 | 6–20 | $1.05 \times 10^{13}$–$3.24 \times 10^{14}$ | 0.61–0.99 | SO$_2$ |
| Fig. 4c3 | | | | |
| Fig. 5a | 5–21 | $6.46 \times 10^{13}$–$4.77 \times 10^{14}$ | 0.42–1.00 | SO$_2$ |
| Fig. 5b | 26–30 | $7.28 \times 10^{13}$–$3.19 \times 10^{14}$ | 0.66–1.00 | CO |
| Fig. 5c | 5–30 | $6.46 \times 10^{13}$–$4.77 \times 10^{14}$ | 0.42–1.00 | SO$_2$ + CO |

The original Tables S3 and S4 were changed to Tables S5 and S6, respectively.

Line 176: "…where $a$–$f$ are fitting parameters (values are reported in Table S5);"

Line 183: "…where $x$–$z$ are fitting parameters (values are reported in Table S6);"

Line 201: "…the parameters $a$–$f$ fitted from our decay experiments (Table S5) should be quite different from those in Li et al. (2015),…"

Lines 202-203: "Similarly, fitting parameters $x$–$z$ for OFR254 mode from our decay experiments (Table S6) are also different from those in Peng et al. (2015)."

Lines 218-219: "…applied in Figure 1a1 is presented in Table S5. When the residence time was increased to 61–200 s,…"

Lines 364-365: "The parameters $x$–$z$ (FP$_{lOHR, 254}$; lOHR: low external OHR) (Table S6) were obtained by fitting Eq. 3 to OH$_{exp, dec}$."

The revisions in Table S3 to Table S5 and Table S4 to Table S6 in the Supplementary Information are not listed here but have been updated in the revised Supplementary Information.

5. *Temperature Effects: Temperature might be an important influencing factor. Field experiments may occur under different locations and seasonal conditions, resulting in significant variation in sample temperature. At the same time, UV lamp pressure directly affects the internal OFR temperature. The reaction rates used in the paper are treated as constants—could this cause significant deviations? Please evaluate this potential impact.*

Response: Thanks for the valuable suggestion. We used low-pressure Hg lamps in this study. We have responded to the comment about the temperature effects in the previous text, referring to the response to reviewer #2 specific comment 8.

6. *Applicability Across Different OFR Designs: Is it feasible to apply the described minimum number of experiments to different types of OFRs? As far as I know, the*

*designs of OFRs can vary significantly. For instance, cone-shaped inlets and premixing inlets can affect residence time distributions within the OFR. Can the $OH_{exp}$ estimation equations be applied universally to these designs? Detailed descriptions are required to clarify this point.*

Response: Thanks for the valuable suggestion. Referring to our response to reviewer #1's comment 1 and reviewer #2's specific comment 21, for oxidation flow reactors other than the PAM-OFR, if the assumption of plug flow conditions is acceptable, then the conclusions stated in our study can hold.

*By addressing these questions, the study can provide a more robust and universally applicable framework for offline $OH_{exp}$ calibration in different experimental and field scenarios.*

**Specific comments:**

1. *L110-L112: In the PAM-OFR system, there are four ultraviolet lamps. Two of these lamps generate ozone, while the other two do not? Regardless of whether the system is operating in OFR185 or OFR254 mode, only two of these lamps are turned on at a time?*

Response:: Thanks for the comment. Yes. Two ozone-producing Hg lamps (GPH436T5VH/4P) were used in the OFR185 mode. They are produced using clear fused quartz to allow transmission of both 185 m and 254 nm. Ozone is then generated via reactions R3-R4 mentioned in the manuscript. Two ozone-free mercury lamps (GPH436T5L/4P) were used in the OFR254 mode. These fused quartz lamps are doped with titanium which blocks the transmission of 185 nm radiation and cannot produce ozone. Whether the system is operating in OFR185 or OFR254 mode, only two of these lamps are turned on at a time.

2. *L114: "quartz tubes" here is the sleeves in L112?*

Response: Thanks for the comment. The sleeve here refers to the layer of quartz sleeve surrounding the UV lamp, which protects the lamp against damage, leakage, temperature fluctuations, etc., without reducing the UV efficiency.

3. *L125: "0.1ppm" here should be 1ppb, ref:*

   *http://www.bjkwnt.com/productinfo/803339.html*

   Response: Thanks for pointing out this mistake. We have corrected it accordingly, which can refer to our response to Reviewer #1's comment 2.

4. *L147: In this paper, are the reaction rate constants of $SO_2$ and CO with OH consistently using a constant value? As can be seen from Figure S2, the temperature inside the reactor varies significantly under different lamp pressures: 23-26°C for the OFR185 mode and 33-36°C for the OFR254 mode. The impact of temperature needs to be assessed, especially for the OFR254 mode.*

   Response: Thanks for the valuable comment. In this study, we use a constant value of the reaction rate constants of $SO_2$ and CO with OH consistently. We have responded to the comment about the temperature effects in the previous text, referring to the response to reviewer #2 specific comment 8.

5. *It is crucial to note that while equation 3 captures the combined effect of light intensity and RH through $r_{O3}$, experiments still need to be conducted at different humidity levels. Could you please inform the readers of the range of RH used in this study?*

   Response: Thanks for the valuable suggestion. We have added a column in Table S2 of the revised Supplementary Information to show the relative humidity involved in the decay experiments conducted in OFR254 mode.

   Supplementary Information:

**Table S2:** List of 37 sets of OFR254 trace-gas decay experiments under different conditions. $SO_2$ or CO was used as the source of $OHR_{ext}$. Each set of experiments was performed under 5–9 lamp intensity settings.

| Experiment ID | Species | Initial concentration (ppb) | $OHR_{ext}$ ($s^{-1}$) | Input $O_3$ concentration (ppm) | Residence time (s) | Water vapor mixing ratio (%) | Relative humidity (%) |
|---|---|---|---|---|---|---|---|
| 1 |  | 286.2 | 6.69 | 4.27 | 69.6 | 1.65 | 27.8 |
| 2 |  | 283.3 | 6.62 | 5.91 | 69.0 | 2.46 | 39.6 |
| 3 |  | 283.8 | 6.63 | 6.17 | 69.0 | 0.99 | 18.1 |
| 4 |  | 289.9 | 6.78 | 6.30 | 69.0 | 1.52 | 28.7 |
| 5 |  | 575.5 | 13.45 | 6.20 | 69.0 | 1.07 | 17.0 |
| 6 |  | 575.2 | 13.44 | 6.08 | 69.0 | 2.45 | 43.8 |
| 7 |  | 583.5 | 13.64 | 6.15 | 69.0 | 1.62 | 29.2 |
| 8 |  | 868.6 | 20.30 | 6.23 | 69.0 | 2.22 | 44.0 |
| 9 |  | 874.7 | 20.45 | 6.05 | 69.0 | 1.57 | 27.7 |
| 10 |  | 868.9 | 20.31 | 6.32 | 69.0 | 0.97 | 16.9 |
| 11 |  | 454.9 | 10.63 | 7.77 | 69.0 | 0.96 | 17.1 |
| 12 |  | 450.2 | 10.52 | 8.16 | 69.0 | 2.17 | 41.8 |
| 13 |  | 450.7 | 10.53 | 6.58 | 69.0 | 1.41 | 28.6 |
| 14 |  | 737.9 | 17.25 | 8.32 | 69.0 | 0.88 | 16.9 |
| 15 | $SO_2$ | 746.8 | 17.46 | 7.77 | 69.0 | 2.20 | 39.9 |
| 16 |  | 747.0 | 17.46 | 9.38 | 69.0 | 1.50 | 26.8 |
| 17 |  | 204.0 | 4.77 | 2.62 | 34.4 | 2.11 | 42.8 |
| 18 |  | 201.9 | 4.72 | 3.00 | 34.4 | 1.32 | 27.9 |
| 19 |  | 196.9 | 4.60 | 2.90 | 34.4 | 0.94 | 19.0 |
| 20 |  | 282.6 | 6.61 | 6.07 | 34.4 | 0.79 | 17.3 |
| 21 |  | 572.9 | 13.39 | 6.11 | 34.4 | 0.82 | 17.6 |
| 22 |  | 908.6 | 21.24 | 6.08 | 34.4 | 0.78 | 17.3 |
| 23 |  | 204.7 | 4.78 | 4.47 | 43.7 | 0.86 | 18.2 |
| 24 |  | 402.8 | 9.41 | 5.47 | 47.4 | 2.14 | 42.0 |
| 25 |  | 459.9 | 10.75 | 5.34 | 54.0 | 1.91 | 36.4 |
| 26 |  | 840.2 | 19.64 | 9.82 | 111.5 | 1.74 | 36.1 |
| 27 |  | 262.7 | 6.14 | 8.23 | 70.9 | 2.27 | 43.0 |
| 28 |  | 430.8 | 10.07 | 8.19 | 70.6 | 2.23 | 45.5 |
| 29 |  | 260.1 | 6.08 | 13.17 | 69.6 | 2.38 | 40.6 |
| 30 |  | 511.4 | 11.95 | 19.39 | 125.5 | 2.61 | 45.2 |
| 31 |  | 4909.4 | 29.02 | 3.15 | 19.8 | 0.91 | 19.0 |
| 32 |  | 4685.6 | 27.70 | 5.06 | 37.4 | 0.83 | 20.8 |
| 33 |  | 4958.3 | 29.31 | 4.13 | 47.1 | 0.78 | 15.9 |
| 34 | CO | 4829.8 | 28.55 | 5.59 | 69.4 | 2.20 | 43.3 |
| 35 |  | 4358.0 | 25.76 | 2.95 | 34.5 | 1.78 | 36.2 |
| 36 |  | 5034.5 | 29.76 | 4.82 | 46.8 | 2.18 | 39.4 |
| 37 |  | 4438.5 | 26.24 | 12.95 | 95.2 | 2.25 | 40.4 |

6. *L161: equation 3: the sign after parameter a should be minus according to (Peng et al., 2015).*

Response: Thanks for the comment. Peng et al. have corrected Eq. 3 on 2 May 2016. The DOI number of corrigendum is *doi:10.5194/amt-8-4863-2015-corrigendum*, where the minus sign after parameter *a* (parameter *a* was changed to parameter *x* in the revised manuscript) is changed to a plus sign. We used the corrected equation according to the corrigendum.

7. *L186-200: This paragraph describes the effect of different residence times on the estimation of $OH_{exp}$, finding that the equation fitting parameters at a residence time of 33s can be applied to longer residence times. However, as seen in Figure 1-a2, estimated $OH_{exp}$ at longer residence times (red points) seems to be higher than the shorter residence times (green points), meaning it deviates further from the 1:1 line. Although this is not obvious in logarithmic coordinates, a quantitative assessment of the error range, as described in L200, should be provided here. Subsequent descriptions on other factors should also include relevant quantitative statements, such as the error magnitude at specific $OH_{exp}$ values.*

Response: Thanks for the suggestion. In Figure 1a2, some green data points (shorter residence time) have higher OH exposure than the data point with the longest residence time. We provided the error for the entire $OH_{exp}$ range involved in our study in our response to general comment 2 of reviewer 3.

8. *Regarding the study on residence time, the range of $OH_{exp}$ is within one day, but the highest $OH_{exp}$ in actual field observations should be at least greater than 3 days. What is the difference between the experiment in Figure S3 and that in Figure 1, and why aren't the data combined?*

Response: Thanks for the comment. Referring to our response to Reviewer #2's specific comment 14, We have modified the display of the photochemical age in Figure

1 to enhance readability. In the section discussing residence time, the range for $OH_{exp,}$

$_{dec}$ in Figure 1a1 is $1.38 \times 10^{10}$–$5.1 \times 10^{11}$ molecules $cm^{-3}$ s, corresponding to a photochemical age of 0.11–3.94 days. For Figure 1a2, the range for $OH_{exp, dec}$ is .38 $\times$

$10^{10}$–$6.45 \times 10^{11}$ molecules $cm^{-3}$ s, corresponding to a photochemical age of 0.11–4.97

days.

Both Figure S3 and Figure 1 aim to investigate whether the fitted parameters *a–f*, obtained within a limited range, are applicable for estimating $OH_{exp}$ when the ranges exceed the original range. However, Figure 1 illustrates the case where this narrow range was situated within the lower interval, with the extended experimental conditions above the original range. In contrast, Figure S3 depicts the case where the narrow range was situated within the higher interval, with the extended experimental conditions falling below the original range. The ranges of each experimental condition for the datasets included in each panel of Figure 1 and Figure S3 are given in Table S3 in the revised Supplementary Information. When selecting data points for single variable research from the 129 $SO_2$ decay experiments, in order to keep the conditions outside of the variable within the same range, some data points in Figures 1 and S3 are different.

They are combined both in Figures 3a and 3c, which include all 129 data points from the $SO_2$ experiments.

*9.  Furthermore, this paragraph only describes the range of residence times and does*

*not detail the values of other experimental variables. Similar descriptions should*

*be included in the subsequent paragraph about other factors.*

Response: Thanks for the valuable suggestion. Referring to our response to comment 3 of the general comment, we have added Tables S3 and S4 in the revised manuscript of the main text and Supplementary Information to show the range of all experimental conditions for the different datasets.

*10. L267-268: Here, the equations obtained by refitting the data for different $OHR_{ext}$*

*ranges show good adaptability. If your experimental conditions are $OHR_{ext} > 23$ s⁻*

*$^1$ but less than 198 s$^{-1}$, it is uncertain whether the fit will still be good, as the sensitivity of those parameters to OHR$_{ext}$ does not appear to be linear.*

Response: We only have limited data points in that range of high OHR$_{ext}$, which prevents good fitting using the empirical equations. Nevertheless, the good agreements for the whole OHR$_{ext}$ suggest that the empirical equations are applicable for most of the OHR$_{ext}$ conditions.

*11. L369: It is mentioned here that the parameters for OFR254 is not as good, but the abstract (L35-37) states that it is not bad. Please maintain consistent descriptions throughout the document.*

Response: Thanks for the valuable suggestion. What we intend to express here is that for the OFR185 and OFR254 models, all parameters obtained within a narrow range of conditions can reliably estimate OH$_{exp}$ when extending the experimental conditions. However, the consistency between OH$_{exp, est}$ and OH$_{exp, dec}$ is better in the OFR185 model. Consequently, we have made the following modifications in the revised manuscript.

Lines 406-407: "…in the estimation equation to estimate OH$_{exp}$. Compared with OFR254 mode, the consistency between OH$_{exp, est}$ and OH$_{exp, dec}$ in OFR185 mode is better. For the OFR254 mode,…".

*Reviewer #4:*

*General comment:*

*Liu et al. re-calibrated the OH exposure ($OH_{exp}$) Empirical Parameterizations with the experiment conditions (OH reactivity (OHR), RH, residence time, $O_3$) range in large scales for OFR185 and OFR254 modes. They found the $OH_{exp}$ from the fitted parameters were less affected by the residence time, water vapor mixing ratio, and $O_3$ concentration. When the OHR was much higher (e.g., more than 200 $s^{-1}$), it would overestimate the $OH_{exp}$ if the parameters were obtained with the data from narrow experiment conditions (e.g., OHR less than 30 $s^{-1}$) for OFR185 mode. They also pointed out that 20-30 data points were needed to obtain reliable parameters. Their results provide a reference for the estimation of $OH_{exp}$ and parameter fitting of OFR in different experiments, such as field observations or source emission experiments.*

Response: We thank the reviewer for the positive comments. We address the questions and comments as follows.

*Specific comments:*

*After reading through the manuscript, I was confused about the aim and novelty of this manuscript:*

1. *Since Rowe et al. 2020 have already published the parameterization results for the lamps produced by light source inc. Why did the authors demonstrate another set of parameters? Are they better compared with Rowe et al. 2020? The comparison with previous study shall be shown here.*

Response: Thanks for the comment. We attempted to directly use the parameter results provided by Rowe et al. (2020) to estimate $OH_{exp}$. When performing linear regression between the $OH_{exp, est}$ and the $OH_{exp, dec}$ calculated through trace gas decay, the slope value was found to be 1.9, which showed an overestimation of OH exposure.
Secondly, the parameterization results from Rowe et al. were derived by fitting empirical equations to the model outcomes. OFR conditions input to the photochemical model are shown in the table below. For the external OHR, the range involved in our study is higher than that in Rowe et al.'s research. The experimental conditions in the PAM-OFR involve not only general atmospheric conditions but also high-$OHR_{ext}$ conditions, e.g., those directly from emission sources. For instance, the $OHR_{ext}$ of direct vehicle emission can be as high as 1000 $s^{-1}$, which has not been reflected in Rowe et al.'s study.

|  | Rowe et al. | This study |
|---|---|---|
| Residence time (s) | 124 | 33–296 |
| Water vapor mixing ratio (%) | 0.1–3 | 0.38–2.76 |
| External OHR of $SO_2$ ($s^{-1}$) | / | 4–204 |
| External OHR of CO ($s^{-1}$) | 0.77–232 | 61–1227 |

Thirdly, from the perspective of practical PAM usage, we would like to know in which ranges the temperature, relative humidity, ozone concentration and other conditions need to be set to achieve the desired $OH_{exp}$ when using PAM-OFR. This aspect was not investigated in Rowe et al. (2020).

*2. The KinSim model can reproduce all the chemistry in the OFR. Do the experiment results done here agree well with the model results? Especially for the high OHR. The comparison of OH exp between the KinSim model and experiments is necessary. especially for the extreme high OHR cases. The parameterization in Rowe et al.2020 was fitted based on the KinSim model output. The model output was constrained by experiment results.*

Response: Thanks for the comment. We did not reproduce all our experimental conditions using KinSim model. Instead, we randomly selected five decay experiments conducted under the OFR185 mode for KinSim simulation. This selection included two $SO_2$ experiments with $OHR_{ext}$ values of 198 $s^{-1}$ and 204 $s^{-1}$, as well as three CO experiments with $OHR_{ext}$ values of 73 $s^{-1}$, 614 $s^{-1}$, and 1126 $s^{-1}$, all of which were conducted under high $OHR_{ext}$ conditions. The linear regression results between the experimental and model results for $OH_{exp}$ yielded an R-squared value of 0.98 and a slope of 1.8. Although KinSim can estimate $OH_{exp}$ by inputting various conditions, from the perspective of PAM-OFR users, we want to know what temperature, relative humidity, ozone concentration, and other parameters are needed to achieve $OH_{exp}$ that we wanted during practical use. Additionally, we fitted the parameterization based on the decay experiment data to obtain parameters that can more accurately estimate $OH_{exp}$.

3. *Line 163, for the species with low $k_{OH}$ e.g., SO$_2$ and CO, the $OH_{exp}$ calculated from plug low might agree with these from residence time. However, the residence time shows substantial influences on the species with high $k_{OH}$. E.g., MT Isoprene (Palm et al., 2018). For these species, I do not think the conclusion shown here will be valid. The same conclusion is also applicable to the section which discussed the residence time influences to the OH exposure.*

Response: Thanks for the valuable comment. To clarify that the plug-flow assumption generally holds only when using species with low $k_{OH}$ (such as $SO_2$ or CO), we have made the following revisions and additions in the revised manuscript:

Lines 162-165: "In the calculation of $OH_{exp, dec}$ (see the paragraph below), plug flow conditions were assumed, which has been shown to agree with the RTD approach for $OH_{exp}$ when using species (such as $SO_2$ or CO) with low reaction rate constants with OH radicals ($k_{i, OH}$) by Li et al. (2015) and Peng et al. (2015)."

Since the definition of reaction rate constants with OH radicals has been given in the revised text above, it will be written directly as $k_{i, OH}$ in lines 166.

Line 166: "…, whose $k_{i, OH}$ have been well characterized…"

We have also made the following revisions and additions in the section which discussed the residence time influences to the OH exposure.

Lines 227-231: "…after taking the logarithm of them. It is important to note that the above discussion regarding residence time assumes a plug-flow condition within the PAM-OFR, which is applicable to substances with low $k_{i, OH}$, such as $SO_2$ (or CO). For species that react rapidly with OH, such as monoterpenes or toluene, localized concentration gradients can develop within the OFR, leading to a significant uneven actual RTD that affects the estimation of $OH_{exp}$ (Palm et al., 2018)."

4. *Lines 128-129: The lamps applied in this study are covered or not?*

    Response: Thanks for the comment. The UV lamps used in this study were partially covered by opaque heat shrink tubing, which reduced the UV light intensity to levels below what is achievable using the ballast dimming voltage when not covered.

5. *Lines 233-235, RH shall also be shown.*

    Response: Thanks for the suggestion. Since the water vapor mixing ratio is a quantity influenced by both temperature and relative humidity, and it appears as a single value in Eq. 1, we only present the $H_2O$ value here. To make this clearer, we have added this part in line 178 as follows:

    Lines 177-178: "$H_2O$ is water vapor mixing ratio in PAM-OFR (%), which is influenced by both temperature and relative humidity;"

    However, for easy comparison with other studies, we also converted it to RH values under the conditions of 101.325 kPa and 298 K, which have been provided in lines 32-35 (but we used the real-time temperature and relative humidity for calculating each water vapor mixing ratio in this study).

    Line 32-35: "For example, parameters derived within a narrow water vapor mixing ratio range (0.49–0.99 %, corresponding to 15.1–30.8 % of relative humidity at 101.325 kPa and 298 K) can be extended to estimate the $OH_{exp}$ under the entire range of water vapor mixing ratios (0.49–2.76 %, equivalent to 15.1–85.7 % of relative humidity under identical conditions)."

6. *Lines 108-111: "Yet, it is unclear whether the fitted parameters obtained under*
    *certain conditions can still accurately estimate $OH_{exp}$ when experimental conditions,*
    *such as UV light intensity, water vapor mixing ratio, residence time, and external*
    *OH reactivity ($OHR_{ext}$), undergo significant changes." I think at least the UV light*

*intensity and water vapor mixing ratio should have been considered in most published papers.*

Response: Thanks for the comment. Many published papers have shown that the variations in UV light intensity and water vapor mixing ratio can affect OH exposure. What we want to express here is that the parameters in empirical equations are generally fitted within specific ranges of conditions. For example, the parameters in Rowe et al.'s study were obtained with the water vapor mixing ratio of 0.03–3.9 % (with other conditions, such as residence time and reactant concentration, also specified). However, we are uncertain whether the original set of parameters can still reliably estimate OH exposure when the conditions that affect $OH_{exp}$ change significantly, such as when the water vapor mixing ratio falls outside the 0.03–3.9 % range.

7. *Line 128: was Š is.*

Response: Thank you for pointing out this mistake. We have corrected the typo.

Line 128: "The OH is generated via OFR185 using two ozone-producing Hg lamps…"

8. *Lines 285-298: The author fitted the parameters based on the equation proposed by Li et al., (2015) and explained the deviation of $OH_{exp}$ under high OHR using the equation. They found the sum of parameters related to OHR first increased and then decreased with the increase of OHR (figure S4a3). The sum of the parameters related to OHR shows a monotonical decreasing trend when using the $FP_{eOHR, 185}$. As the OHR from $SO_2$ only reaches 204 $s^{-1}$, will the curve in Figure S4b3 increase at higher OHR, as the one shown in Figure S4a3 (or what is the range of OHR applicable to $FP_{eOHR, 185}$)?*

Response: Thanks for the comment. Using $FP_{eOHR, 185}$, when the sensitivity test maintains the conditions described in lines 284-285 and only the $OHR_{ext}$ value is increased, the curve in Figure S4b3 continues to decrease monotonically at higher $OHR_{ext}$ values (at least monotonically decreasing up to the 2000 $s^{-1}$ that we detected).

[revised manuscript text omitted]

---

## Author Response (AR2)

Response to Editor

We thank the editor for handling this manuscript and providing constructive comments. We provide herein response and make changes according to the editor's comments.

Public justification (visible to the public if the article is accepted and published):

1. Add response to comment 8 (line 123) into the manuscript.

Response: we have in this version included a brief discussion on the potential effect of temperature change at the end of section 2.2, as follows.

Despite the use of nitrogen as a purge gas to reduce the heat generated by the lamp, temperature variations were still observed within the PAM-OFR. There was a maximum deviation of approximately 13 °C from 25 °C when using $SO_2$ as the OHR source. However, the $k_{SO2, OH}$ was $8.85 \times 10^{-13}$ $cm^3$ molecule$^{-1}$ s$^{-1}$ at 38 °C (Burkholder et al., 2020), which results in the calculated $OH_{exp, dec}$ being only approximately 7% higher than that derived from $k_{SO2, OH}$ at 25 °C. Pan et al. (2023) noted that temperature increases caused by lamp heating exerted minimal influence on gas-phase reaction rates, with $SO_2$ decay and OH exposure showing negligible variations. Therefore, the influence of temperature on reaction kinetics was not considered in this study.

2. Add response to comment 13 into the manuscript.

Response: we have in this version included the reason why error bars were not included in Figure 1, as follows.

All the error values for $OH_{exp, dec}$ are half or even two orders of magnitude smaller than the corresponding $OH_{exp, dec}$ values. When applying a logarithmic scale, the error bars become difficult to represent. To enhance the readability of the graph, error bars have not been included. For the same reason, error bars for $OH_{exp, est}$ values are also not displayed.